# Flexopiezoelectricity at ferroelastic domain walls in WO$_3$ films

Shinhee Yun [1,6], Kyung Song [2,6], Kanghyun Chu[1,3,6], Soo-Yoon Hwang [4], Gi-Yeop Kim[4], Jeongdae Seo[1], Chang-Su Woo[1], Si-Young Choi [4✉] & Chan-Ho Yang [1,5✉]

The emergence of a domain wall property that is forbidden by symmetry in bulk can offer unforeseen opportunities for nanoscale low-dimensional functionalities in ferroic materials. Here, we report that the piezoelectric response is greatly enhanced in the ferroelastic domain walls of centrosymmetric tungsten trioxide thin films due to a large strain gradient of $10^6 \, \mathrm{m}^{-1}$, which exists over a rather wide width (~20 nm) of the wall. The interrelationship between the strain gradient, electric polarity, and the electromechanical property is scrutinized by detecting of the lattice distortion using atomic scale strain analysis, and also by detecting the depolarized electric field using differential phase contrast technique. We further demonstrate that the domain walls can be manipulated and aligned in specific directions deterministically using a scanning tip, which produces a surficial strain gradient. Our findings provide the comprehensive observation of a flexopiezoelectric phenomenon that is artificially controlled by externally induced strain gradients.

[1] Department of Physics & Center for Lattice Defectronics, Korea Advanced Institute of Science and Technology (KAIST), Daejeon 34141, Republic of Korea. [2] Department of Materials Analysis and Evaluation, Korea Institute of Materials Science (KIMS), Changwon 51508, Republic of Korea. [3] Group for Ferroelectrics and Functional Oxides, Institute of Materials, Swiss Federal Institute of Technology in Lausanne (EPFL), 1015 Lausanne, Switzerland. [4] Department of Materials Science and Engineering, Pohang University of Science and Technology (POSTECH), Pohang 37673, Republic of Korea. [5] KAIST Institute for the NanoCentury, KAIST, Daejeon 34141, Republic of Korea. [6]These authors contributed equally: Shinhee Yun, Kyung Song, Kanghyun Chu. ✉email: youngchoi@postech.ac.kr; chyang@kaist.ac.kr

Ferroelasticity is a mechanoelastic phenomenon in which a crystalline lattice spontaneously deforms[1]. The manifestations of ferroelasticity include superelasticity, shape memory, and diffusionless martensitic transformation[1,2]. Since the resulting strain field is not spatially uniform, the strain gradient—the rate at which the strain changes through a position—gives rise to flexoelectricity[3,4]. On a macroscopic scale the polarization induced by flexoelectricity is typically negligible due to the low strain gradients, of ~1 m$^{-1}$. However, nanoscale materials and devices can tolerate strain gradients as large as $10^6$ m$^{-1}$, and this can lead to a substantial amount of flexoelectric polarization which may be comparable to or even larger than conventional ferroelectric polarizations[5–7].

Under these conditions, the walls or boundaries between two crystallographically different domains[8,9] can undergo a large amount of lattice deformation and/or structural reconstruction. Because of this feature, ferroelastic domain walls in centrosymmetric, nonpolar materials have attracted much attention not only for their influence on conductivity[10] and superconductivity[11] but also for interfacial polarity phenomena, such as the nonlinear optical effect[12–14] and field-induced gating effect on electronic conduction[15,16]. Recent theoretical studies have also shown the importance of antiferrodistortive octahedral rotations and tilts and polar displacements at ferroelastic domain walls[17–19]. The interplay of complex ferroelastic twin patterns and flexoelectrical polarity has been theoretically studied in terms of kink and junction generation reminiscent of polar vortices[20].

One of the most promising ways of visualizing the polar property at the domain walls is to utilize the piezoresponse based on converse piezoelectricity. However, this observation has rarely been made[21] because most ferroelastic domain walls are too narrow in width[22,23] or undergo atomic reconstruction, generating unique interfacial properties that are totally different from the bulk[2,24]. Therefore, although a piezoresponse is expected to be present in the ferroelastic domain walls upon symmetry breaking, the direct detection of a strain gradient-induced piezoresponse at the domain walls still remains a tantalizing challenge.

To visualize the piezoresponse at the ferroelastic domain walls with large strain gradients, the so-called flexopiezoelectricity, we choose epitaxially grown tungsten trioxide (WO$_3$)[25,26], an extensively studied ferroelastic material that shows periodically ordered domain walls in a herringbone structure, where the intervals between the domain walls can be scaled simply by film thickness[25]. This binary oxide looks like an A-site vacant perovskite structure, and thus, WO$_3$ is elastically flexible, as supported by experimental and theoretical results. These include a small Young's modulus and shear modulus[27,28], polaronic transport[29], and various temperature-dependent structural phase transitions caused by phonon softening[30,31]. Accordingly, the domain walls in WO$_3$ provide an ideal platform to explore the possibility of flexopiezoelectricity.

## Results

### Ferroelastic domain pattern in the monoclinic WO$_3$ film. The typical WO$_3$ herringbone pattern, shown in a topographic image (Fig. 1a), consists of dense stripes aligned along the crystalline axes ([1$\bar{1}$0]$_{YAO}$ or [001]$_{YAO}$; the subscript 'YAO' represents the crystal axes of the YAlO$_3$ substrate in the orthorhombic index) at intervals of a few 10 nm. These fine stripes, actually fine-domain walls, are attributed to the surface slope changes at the twin boundaries between monoclinic A$_1$ and A$_2$ ferroelastic domains or between B$_1$ and B$_2$ ones. Mosaic rotations occur alternatingly in a clockwise or counterclockwise manner, as schematically shown in Fig. 1a. On a larger length scale (a few 100 nm), we can identify other stripes, the so-called macro-domain walls, which

are rotated by 45° relative to the fine-domain walls, as indicated by the dotted line in Fig. 1a.

The ferroelastic twin structure is apparently seen in a cross-sectional weak-beam dark-field transmission electron microscopy (TEM) image taken along the zone axes of [001]$_{YAO}$ and [1$\bar{1}$0]$_{YAO}$ (Fig. 1b–d). TEM images of the [001]$_{YAO}$ zone axis (Fig. 1b, c and Supplementary Fig. 1 in Supplementary Note 1) show that the fine-domain walls appear in the regions of the B domains, but not in the A domains due to the orthogonal relation of the fine-domain walls. Accordingly, one way of distinguishing the A and B domains is to determine whether the fine-domain walls appear in a particular zone axis.

From the [1$\bar{1}$0]$_{YAO}$ zone axis, selected area electron diffraction (SAED) patterns were obtained for the areas of the A and B domains (Fig. 1e, f, respectively). The SAED patterns can be analyzed based on the monoclinic structural phase γ-WO$_3$ (space group P2$_1$/n). A simulation of the electron diffraction patterns depending on the zone axis clearly agreed with the experimental results (Supplementary Fig. 2 in Supplementary Note 1). Our epitaxial WO$_3$ film demonstrates a unique way of relaxing the interfacial strain between the WO$_3$ film and the YAlO$_3$ substrate by alternating the appearance of the shorter a-parameter 3.66 Å and the longer b-parameter 3.76 Å in WO$_3$, compared to the pseudocubic lattice parameters of YAlO$_3$, 3.68 and 3.71 Å (ref. [25]). It is worth mentioning that the hierarchical twin structure appeared as a metastable (sustainable itself) structure compatible with the substrate, not an elastically deformed state forced by strong substrate clamping. We have often observed that the domain structure remains in TEM specimens in areas where the substrate has been removed.

### Lateral piezoresponses at ferroelastic domain walls. We performed angle-resolved lateral piezoresponse force microscopy (PFM) over two crystallographically distinguished domains indicated by 'A' and 'B' and their boundaries. We expect that the two neighboring domains are coherently connected to each other, thereby leading to a large strain gradient at the interface (Fig. 2a). We also note that the strain gradients of the two adjacent domain walls have opposite signs. In the following descriptions of strain gradients, we will use two Cartesian coordinates; (x, y, z) whose basis axes are parallel to the axes of the pseudocubic WO$_3$ unit cell, and (x′, y′, z′) where the y′-axis (x′-axis) is parallel (perpendicular) to the domain wall along [110]. It is worth noting that the shear strain gradient $\frac{\partial \varepsilon_{x'y'}}{\partial x'}$ remains significant in the (x′, y′, z′) coordinate (see Supplementary Note 2).

We acquired four different lateral PFM images with different tip orientations (Fig. 2b, c) over an identical area of a 200-nm-thick WO$_3$ thin film. The piezoresponse signals along lines 1 and 2 show a tip-orientation dependence (Fig. 2d). High-resolution angle-resolved lateral PFM analyses were performed on the two representative regions, leading to direct visualization of the piezoresponse distribution (Fig. 2e)[32,33]. The amplitudes of the lateral piezoresponses are significantly large at the domain walls compared to the out-of-plane piezoresponses (Supplementary Note 3). The magnitude of the lateral piezoresponse is as much as ~6 pm V$^{-1}$, which is comparable to the typical value observed in epitaxial BiFeO$_3$ films[32]. Their directions are parallel to the ferroelastic domain walls, and the piezoresponse vectors at the neighboring walls are in the opposite directions. This antiparallel arrangement is related to the in-plane shear strain gradients at the domain walls.

Meanwhile, our domain walls deduced from the monoclinic WO$_3$ behaved differently than those reported in tetragonal WO$_{3-x}$ (ref. [10]), where the surface piezoelectricity was reduced at the domain walls along with an increase in conductivity. In

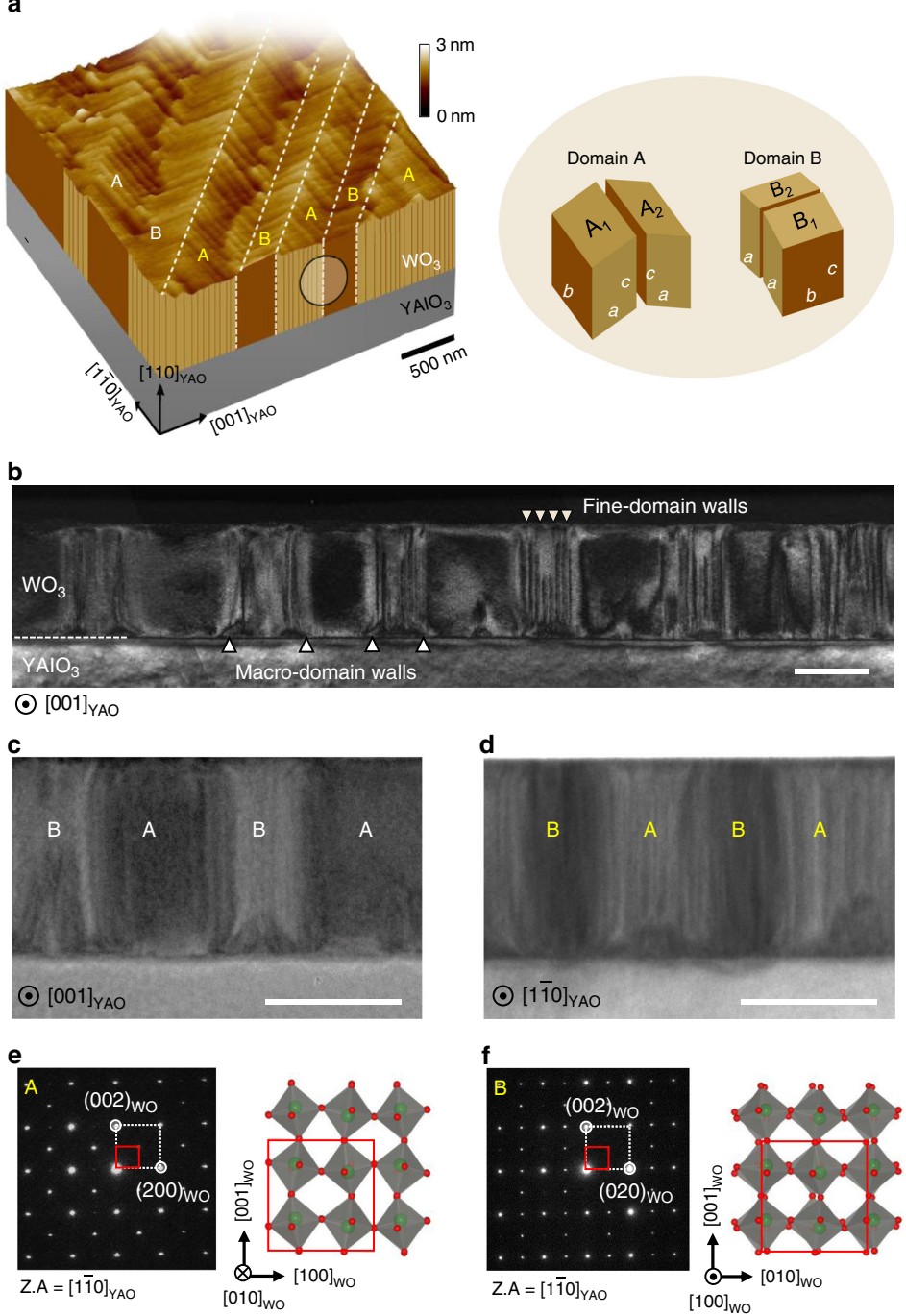

**Fig. 1 Ferroelastic domain structure of epitaxial WO₃ films on YAlO₃ substrates. a** The surface morphology of a 610-nm-thick film with four-oriented monoclinic unit cells. The white dashed lines represent the macro-domain walls. The nearby crooked lines are step edges with a single unit-cell height indicating the films were grown in the step-flow mode. **b** The cross-sectional TEM (dark field) image taken with reflection $g = [00\bar{2}]_{WO}$ shows the alternating A and B domains with fine- and macro-domain walls in a ~300-nm-thick film. **c, d** Zoomed-in bright-field images along the $[001]_{YAO}$ (**c**) and $[1\bar{1}0]_{YAO}$ (**d**) zone axes. **e, f** SAED patterns of the A and B domains, taken along the $[1\bar{1}0]_{YAO}$ zone axis. A white (or red) box represents the pseudocubic unit cell (or monoclinic unit cell with eight octahedrons). The monoclinic unit cells in real space are shown and the subscript '$_{WO}$' represents the crystal axes in the monoclinic cell. From a crystallographic viewpoint, we categorize the macro-domains into A or B domains, depending on whether the $[010]_{WO}$ axis (with lattice constant $b$) of the monoclinic unit cell is parallel to the substrate $[1\bar{1}0]_{YAO}$ or $[001]_{YAO}$. Scale bars indicate 200 nm.

contrast, no enhanced electronic conduction was detected at any macro-domain wall in our study. This discrepancy is presumed to be due to the direction of electric polarization and the relevant surface screening charge. For example, if the flexoelectric polarization was oriented along the out-of-plane direction in

our monoclinic WO₃, electronic carriers must have been induced on the surface suppressing the piezoresponses.

**Origin of flexopiezoelectricity in WO₃.** The piezoelectric response can be loosely understood to occur when a local lattice

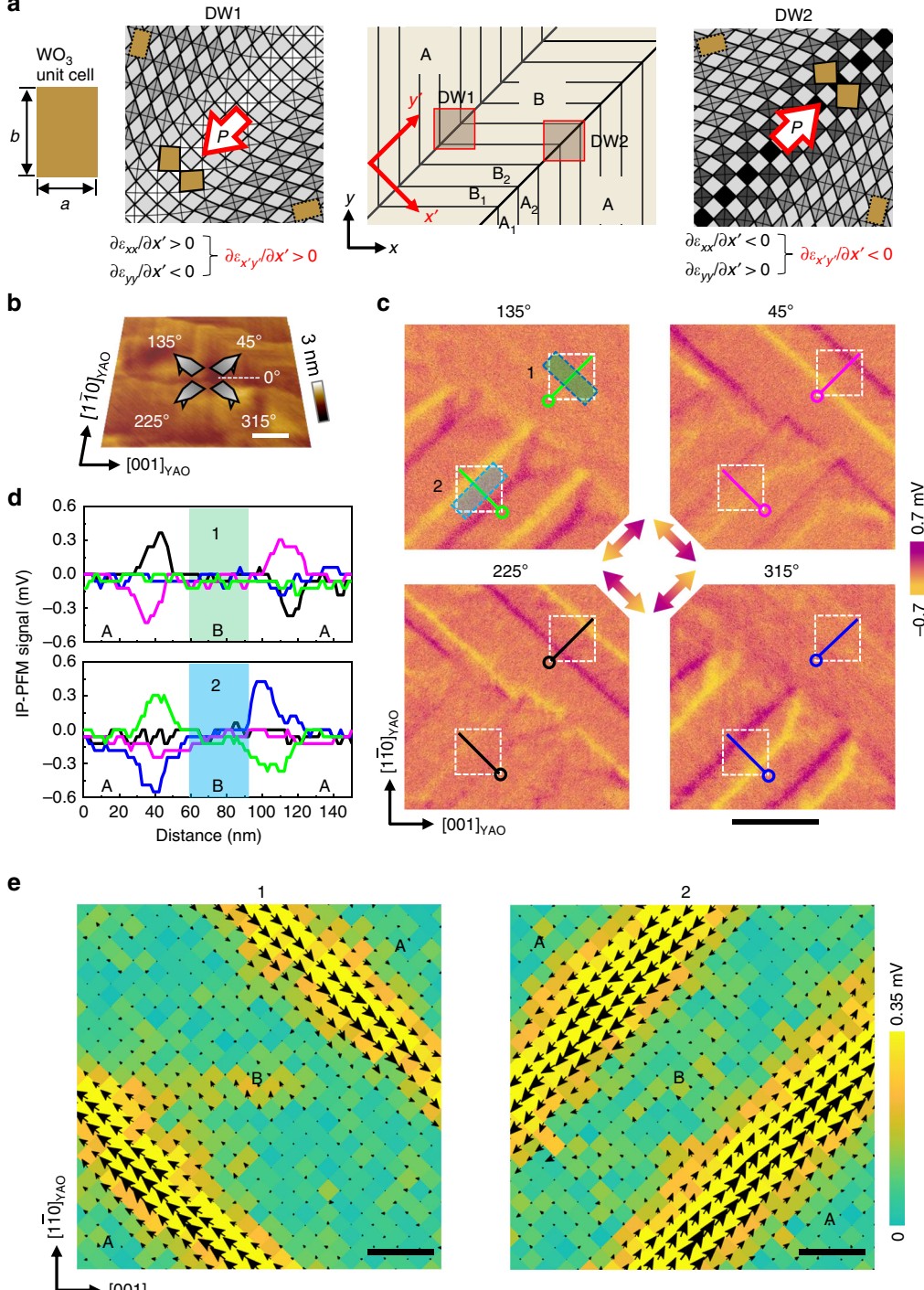

**Fig. 2 Strain-gradients and lateral piezoresponses at the domain walls. a** Schematic of the shear strain-gradients at the macro-domain walls and their piezoresponses. A Cartesian coordinate system ($x'y'z'$) is defined so that the $y'$-axis is parallel to the walls. **b** Schematic of the tip orientations for the PFM measurements with the surface image. **c** Real part signal images of the in-plane PFM for four different cantilever orientations. The color scale represents the direction and magnitude of the in-plane PFM signal as indicated by the central double arrow per panel. Horizontal scale bars in **b**, **c** represent 200 nm. **d** Line profiles of the real part signals along the lines (starting at the circles) in **c**. While a meaningful signal of the piezoresponse is not observed in the interior domains, indicated by the pale green and cyan rectangles, due to the centrosymmetric behavior of WO₃, the ferroelastic domain walls between the A and B domains reveal largely enhanced piezoresponses. Black, blue, magenta, and green colors represent the cantilever orientation angles. 1 mV is estimated to ~30 pm. **e** Angle-resolved lateral PFM maps for the white dashed boxes in **c**. Black arrows represent the in-plane piezoresponse vectors, which are comprehensively obtained by sinusoidal fittings of the real part signals. The color scale represents the magnitude of the arrow. Horizontal scale bars represent 20 nm.

deformation with a strain gradient breaks the inversion symmetry, thereby making the local structure piezoelectrically active. In a stricter sense, the piezoelectric response at the domain wall is necessarily described by a phenomenological trilinear term related to the order parameters of the strain ($\varepsilon_{kl}$), strain gradient ($\frac{\partial \varepsilon_{mn}}{\partial x_j}$), and polarization ($P_i$) of the free energy density of the undeformed parent material,

$$F_{\text{flexoPE}}(P, \varepsilon) = -\sum_{ijklmn} \theta^{\text{f}}_{ijklmn} \delta\varepsilon_{kl} \frac{\partial \varepsilon_{mn}}{\partial x_j} \delta P_i, \quad (1)$$

where all the summation subscripts stand for the spatial coordinate axes. The proportional coefficients, hereafter named by the intrinsic flexopiezoelectric coefficients, are denoted by a six-rank tensor ($\theta^{\text{f}}_{ijklmn}$), and accordingly, flexopiezoelectricity is omnipresent in the crystalline solids irrespective of the existence of the inversion symmetry. Based on this term, the converse piezoelectric response can emerge at the domain walls even in the centrosymmetric material. The effective converse piezoelectric coefficient due to a strain gradient can be obtained through free energy minimization (see Supplementary Note 4):

$$d^{\text{f}}_{abp} = \frac{\delta\varepsilon_{ab}}{\delta E_p} \approx \sum_{ijklmn} \theta^{\text{f}}_{ijklmn} C^{-1}_{abkl} \frac{\partial \varepsilon_{mn}}{\partial x_j} \epsilon_0 \chi_{ip}, \quad (2)$$

where $C^{-1}_{abkl}$, $\epsilon_0$, and $\chi_{ip}$ represent the elastic compliance tensor, vacuum permittivity, and electric susceptibility tensor, respectively. According to the previous PFM result, $d^{\text{f}}_{y'y'y'}$ is expected to be significant.

The flexopiezoelectric coefficient can be estimated by comparing the piezoelectric response and the strain gradient. Accordingly, we directly investigated the strain gradient across the domain walls using atomic scale plane-view scanning TEM (STEM) measurements (Fig. 3 and Supplementary Note 5). In the monoclinic unit cell there is a difference in the pseudocubic lattice parameters $a$ and $b$. The WO$_3$ has spontaneously nonzero lateral longitudinal strains ($\varepsilon_{xx}$ and $\varepsilon_{yy}$) which are defined as relative

deviations from the reference cubic cell. The A and B domains can be transformed into each other through azimuthal rotation around the $c$-axis by $\pm 90°$ (Supplementary Fig. 10). It is remarkable that the crystal structure is not broken regardless of the extremely small curvature of bending at a few 10-nm scale, which was proven by the detailed atomic arrangement using high-resolution STEM.

Usually, ferroelastic domain walls exhibit an abrupt structural change at the atomic scale and/or undergo structural reconstructions, resulting in a failure of the phenomenological continuum description using a strain gradient. However, the twin-type domain walls observed in these WO$_3$ thin films had a wide width of ~20 nm or larger, without significant dislocations (Fig. 3a and Supplementary Fig. 12). Using X-ray diffraction for the HK reciprocal space of the (001) reflection, we observed a diffusive and broad-peak feature along the <110> directions, which are attributed to the domain wall regions. By evaluating the correlation length of the diffusive peak in a nondestructive manner, we were also able to estimate the domain wall width to be ~26.5 nm, which is very close to the value of the STEM measurement (Supplementary Note 6).

Based on direct measurement of the unit-cell deformations between the A and B domains (Fig. 3b), the strain, $\varepsilon_{xx}$ (or $\varepsilon_{yy}$), decreased (or increased) by ~1.5% along the line profile perpendicular to the domain wall within the analysis range (see Supplementary Fig. 13 for analysis on a larger scale). The strain gradients at the center of the domain wall were measured to be ~$\pm 10^6$ m$^{-1}$ (Fig. 3c). These values are comparable to the predicted strain gradient (Supplementary Note 2).

On these grounds, the intrinsic flexopiezoelectric (which can be called flexopiezovoltage) coefficient ($\theta^{\text{f}}_{ijklmn}$) was estimated to be ~30 J C$^{-1}$ [V], by dividing the measured effective converse piezoelectric coefficient (~6 pm V$^{-1}$) by the elastic compliance tensor (~$4.1 \times 10^{-12}$ m$^2$ N$^{-1}$, refs. [27,28]), strain gradient (~$10^6$ m$^{-1}$), vacuum permittivity (~$8.85 \times 10^{-12}$ C V$^{-1}$ m$^{-1}$), and electric susceptibility (~5000, ref. [34]) (see Supplementary Note 4).

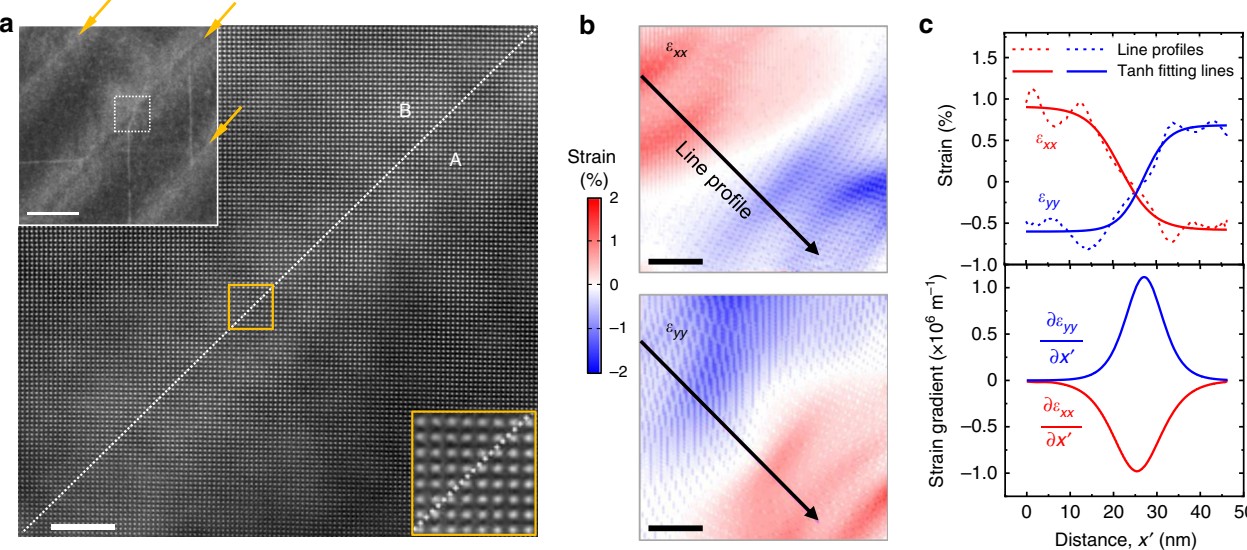

**Fig. 3 Direct observation of the strain-gradients at the ferroelastic domain walls. a** High-resolution scanning TEM images of the domain walls coherently connecting neighboring domains without producing dislocation. Main image is an enlarged image of the white dashed box in the low magnification image in the top left inset. The bottom right inset is the enlarged image in the orange box. Scale bars in the main figure and the inset represent 5 and 50 nm, respectively. **b** Strain analysis for the region of **a**. Scale bars represent 10 nm. **c** Line profile (top) and its derivative plot (bottom) along the $x'$ direction. The left-top inset in **a** with a lower magnification shows the three domain walls indicated by the arrows. The brighter contrasts along the domain walls result from the larger number of scattered electrons due to the dechannelled atomic columns, which is strengthened by a variation of the strain and accordingly the remarkable strain gradient is shown in **b** and **c**.

**Origin of flexoelectricity in WO₃.** To further ensure the existence of strain gradients at the domain walls, we examined the flexoelectric polarization, because flexoelectricity is a universal phenomenon present in all crystalline solids subjected to strain gradients. The flexoelectric polarization ($P_i^f$) depends on the strain gradients through the phenomenological relation:

$$P_i^f = \sum_{jkl} \mu_{ijkl} \frac{\partial \varepsilon_{kl}}{\partial x_j}, \qquad (3)$$

where $\mu_{ijkl}$ are the fourth rank tensor components, called flexoelectric coefficients. The shear flexoelectric coefficient ($\mu_{y'x'y'x'}$) is assumed to be the approximate value of negative $\sim 6.6 \times 10^{-8}$ C m$^{-1}$, which is obtained by multiplying the electric permittivity of WO₃ ($\sim 5000\epsilon_0$) and the intrinsic 'flexovoltage' coefficient of a perovskite oxide ($-1.50$ V) (ref. [35]). In this way, the induced flexoelectric polarization was roughly estimated to be $\sim 13$ μC cm$^{-2}$ (see Supplementary Note 7 for details).

We were able to successfully stabilize the hierarchical domain structure of WO₃ in a phase field simulation using our phenomenological free energy model, containing the electric, elastic, flexoelectric, flexopiezoelectric, and Ginzburg gradient energy terms (see Supplementary Note 8 for details). In this way we were able to visualize the strains, strain gradients, and the electric polarizations over the domains and across the domain walls. The simulated wall width and electric polarization at a macro-domain wall (Supplementary Fig. 17) showed good agreement with the semiquantitative estimates. The fact that the domain structure could be stabilized without the elastic constraint from the substrate suggests the herringbone domain structure is a metastable state, chosen because of good matching with the substrate, with minimal deformation.

The presence of a polar structure leads to a depolarization field. To directly observe in-plane depolarized fields, a plane-view TEM specimen was prepared by polishing away the substrate from the back side (top view in Fig. 4a and Supplementary Fig. 9). We were able to reconstruct the vector map of the electric fields (Fig. 4b) using the differential phase contrast (DPC) STEM technique[36–38]. As expected, the depolarized fields of the flexoelectric polarization induced by opposite shear strain gradients in neighbors were clearly detected at the domain walls. We also executed cross-sectional DPC-STEM measurements (side view in Fig. 4a). We obtained the $x$ and $z$ components of the electric fields inherent in the twin structure (Fig. 4c). By combining them, we constructed a vector map of the electric fields (Fig. 4d). Notably strong electric fields are observed at the ferroelastic domain walls. The directions of the depolarized fields were in contrast to the piezoresponse vectors sensed by the lateral PFM measurements. Because the domain walls are inclined to the zone axis by 45°, the expected electric fields at the walls are projected onto the cross-sectional plane (see Supplementary Note 9 for more details).

Although the flexoelectric polarization is a well-known and universal property in dielectric materials, our observation of the piezoresponse at a domain wall bearing a strain gradient is a hidden phenomenon. It is worth noting that the relative ratio for the measured flexopiezoelectric response to the flexoelectric polarization normalized by the elastic compliance is $\sim 10$, which is of the same order of magnitude as the ratio of the intrinsic flexopiezoelectric constant (30 V) to the intrinsic flexoelectric constant (a few V, ref. [35]).

**Mechanical alignment of ferroelastic domain walls.** Last, we demonstrated that the flexopiezoelectric vector at the ferroelastic domain wall can be manipulated by applying a mechanical stress gradient. We used the trailing effect of an atomic force microscopy (AFM) tip[39] to align the domain walls. A stationary tip exerts electric fields and mechanical stresses isotropically around it. However, as the tip moves, the frictional stresses on the sample surface cause shear mechanical deformation on both sides of the moving trail. These shear stresses are expected to have opposite signs, thereby offering a useful pathway for the local and controllable generation of a shear stress gradient. It was interesting to investigate the use of tip-driven shear stress gradients as a method of controlling the ferroelastic domain walls (Fig. 5a).

To understand the shear strain gradients induced by a frictional force, we performed finite-element modeling (FEM) (details in the 'Methods' section). The shear strains were maximized right next to the contact area (dashed circle) and had opposite signs on the left and right sides of the tip trail. The maximum value ($\sim 0.025$) of the induced shear strain was larger than that of the static twin domains ($\sim 0.013$), which means that the induced shear strains are enough to switch the domains and consequently the domain walls.

Based on this effect, we executed three consecutive scans with a normal force of $\sim 100$ nN (Fig. 5b). The kinetic frictional force was estimated to be $\sim 40$ nN (= the coefficient of kinetic friction × normal force = $\sim 0.4 \times \sim 100$ nN). Interestingly, irregularly oriented ferroelastic domains (patches of 45° or $-45°$ stripe domains) were aligned after a few repeated scans of an AFM tip (see the lateral PFM images taken before and after the switches in Fig. 5c, d). The switching areas remained for more than 30 days. The applied $\sim 100$-nN-normal force is quite a bit lower than those for ferroelectric switching (a few μN, ref. [40]). Reversible switching needs to be emphasized because it implies the lack of linear or planar defects, which leads to wall pinning during the alignment of the domain walls.

## Discussion

We also questioned why the macro-domain wall has a wide width while the fine-domain wall appears as a sharp interface. The fine-domains form in such a way that the monocline $c$-axis is aligned normal to the substrate by mosaic rotation. The facing $bc$ surfaces of two neighboring fine-domains (either A₁ and A₂ or B₁ and B₂) show good compatibility with each other. So, the sharp interface of two crystals can be energetically stable. On the other hand, the macro-domain wall is at the boundary of the two crystallographically different A- and B-domains, where the in-plane crystal axes are transformed into each other by an azimuthal rotation of $+-90°$. The difference between the $a$- and $b$-axis lattice parameters induces a local deformation of unit cells at an $\{110\}_{\rm WO}$-type interface even if the compatibility relation for a symmetric W wall is satisfied. In this situation, a gradual wide lattice deformation occurs in the soft material. If a material is sufficiently rigid elastically, the elastic energy loss is too high to take the deformation. Instead, it prefers to produce other surface reconstructions alongside defect formation, within a relatively narrow interfacial region. This raises a basic question—which physical parameters quantitatively determine the width of the domain wall?

Domain wall widths are determined by competition between an on-site energy and an inter-site energy (e.g., magnetic anisotropy energy versus magnetic exchange energy in magnetic systems). In this ferroelastic system, elastic energy such as the Hook's law with a quadratic dependence on strain plays the role of on-site energy, while the quadratic dependence of strain gradient acts as an inter-site interaction. The smaller the elastic energy compared to the gradient energy, the wider the domain wall width. Our phase field simulation also supports the importance of the Ginzburg gradient energy in controlling the domain wall width (Supplementary Note 8). This would be close in spirit to the work of Salje et al., where sharp or smooth ferroelastic

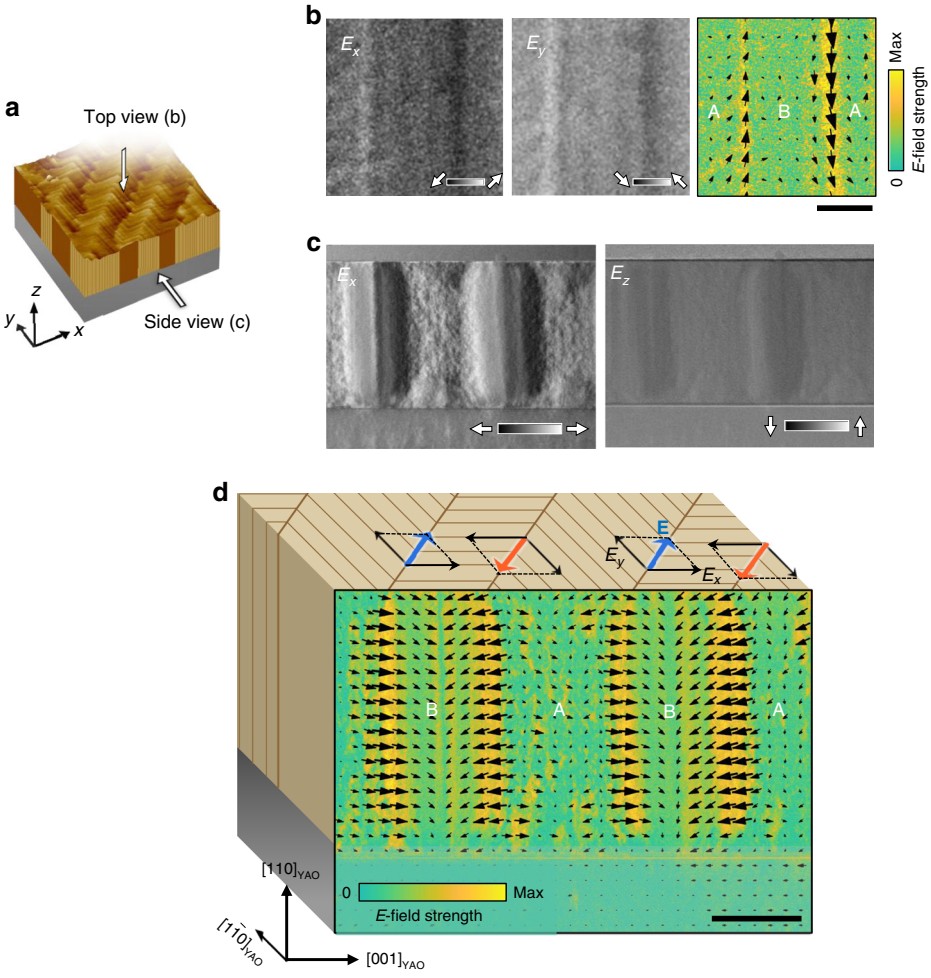

**Fig. 4 Electric fields at the polar domain walls, constructed using the DPC of a transmission electron beam. a** Schematic of the zone axes for the DPC TEM measurements. **b** For the in-plane view ($[\bar{1}10]_{YAO}$ zone axis), contrasts of $E_x$ (left) and $E_y$ (middle) are shown on the same scale for an identical area. Right image is a vector map created by combining $E_x$ and $E_y$. **c** $E_x$ (left) and $E_z$ (right) contrasts for an identical area of the cross-sectional view ($[1\bar{1}0]_{YAO}$ zone axis) on the same scale. **d** Its vector map on the cross-sectional surface. Blue and red arrows on the schematic of the film surface represent the depolarized fields associated with the induced polarizations at the domain walls. Their horizontal components ($E_x$) strongly contribute to the DPC measurement. The domain regions show relatively small signals; however, we can still recognize different electric field configurations in the A and B domains; the A domains look more fuzzy in this geometry of the zone axis $[1\bar{1}0]_{YAO}$. Black scale bars represent 100 nm.

interfaces were theoretically obtained in stiff or soft ferroelastic lattices[20].

In the macro-domain wall region, the shear strain evolves from a positive value to the opposite negative value across the domain wall. The local strain and strain gradient are mainly relevant to the in-plane axes, i.e., the bending structures within the in-plane are just stacked along the normal of the film. So, we guess that the macro-domain wall width has little dependence on film thickness. This is quite distinct from the typical macro-domain's width ($w_m$), which increases with film thickness ($t$) according to the scaling of a quasi-Kittel's law ($w_m \propto t^{0.6}$) (ref. [25]). We performed additional PFM characterization of several samples with different film thicknesses (Supplementary Note 10). Compared to the domain width, the macro-domain wall width had almost an identical value (~30 nm). As a result, the surface areal fraction of the domain wall regions increased as the film thickness became thinner.

We unraveled a little-explored type of electromechanical response in the centrosymmetric tungsten trioxide. The piezoresponse in our epitaxial, monoclinic $WO_3$ was not like conventional piezoelectricity, but was found to be enhanced in proportion to the strain gradient. The intrinsic flexopiezovoltage

coefficient, ~30 V in our study enables the flexopiezoelectric response to be estimated in any given similar class material by multiplying the intrinsic coefficient by elastic compliance, dielectric constant, and strain gradient.

Flexopiezoelectricity is not limited to a specific material but is a universal phenomenon in nanoscale materials/interfaces subject to strain gradients. The flexopiezoelectric coefficient is represented by a sixth-order (even-ranked) tensor, so the effect can be nonzero regardless of the presence or absence of inversion symmetry. These findings will have a broad impact on nanoscale materials and device research. For example, corrugated 2D materials have significant transverse strain gradients and, as a result, the piezoelectric response should appear, according to this knowledge. We expect the piezoelectric response can be generally induced by wrinkling or bending any flexible material.

$WO_3$ is a representative example where the flexopiezoelectricity is significantly manifested, because it has a large dielectric constant (~5000, ref. [34]) and it is also elastically soft[27,28]. (A large flexopiezoelectric effect at the same intrinsic coefficient can emerge as a result). More interestingly, many unusual phenomena can originate from A-site vacant perovskites, such as fast ionic penetration/migration[41], polaronic conduction[29], superconductivity[11], and many

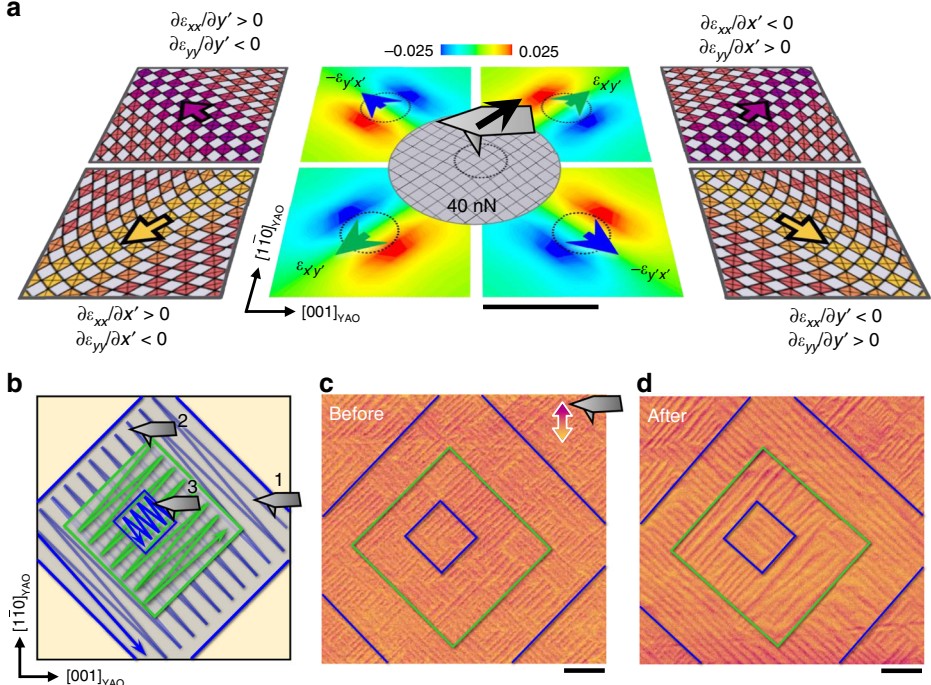

**Fig. 5 Mechanical switching of the domain walls by an AFM tip. a** The middle image shows the FEM-simulated shear strain distribution induced by the trailing mechanical force depending on the tip direction (blue and green arrows). Black scale bar represents 50 nm. The dashed circle represents the contact area of an AFM tip. Left and right schematics show the lattice deformations and flexoelectric polarizations of four-variant domain walls. The colors of the arrows indicate the directions of in-plane PFM signals when acquired at the tip orientation in **c**. **b** Three consecutive switching through over-layered scans with a frictional force of ~40 nN. Blue and green lines indicate the abbreviated trajectories of a tip along −45° and 45°. The number represents the order of switching. **c, d** Lateral PFM images of the WO₃ domain walls before and after the mechanical switching. The color contrasts (purple and yellow) indicate the macro-domain walls shown in the same arrow color in the outer schematics in **a**. Scale bars represent 1 µm. Compared with the as-grown region with denser domain walls and irregularity, the switched areas subjected to frictional forces that align the macro-domain walls have more uniform stripes inside the switched boxes.

polymorphic phases as well as soft modes[30,31], can be further coupled to the flexopiezoelectricity. From a microscopic point of view, it is also interesting to clarify that the A-site vacant space also acts as a buffer space, capable of enduring a large elastic deformation in the future.

Consequently, WO₃ is a helpful example for understanding the high-order effects of piezoelectricity or flexoelectricity. It offers a phenomenological base for answering quantitatively and symmetry-wise how piezoelectric coefficients can be modified by strain gradients, or how flexoelectric coefficients can be changed by strain. Just as the value of the flexovoltage coefficient is universally valid for many perovskite oxides[35], so is the estimate of the intrinsic flexopiezoelectric coefficient (i.e., the flexopiezovoltage coefficient).

We have revealed the existence of flexopiezoelectricity at ferroelastic domain walls using an integrative approach, employing high-resolution angle-resolved lateral PFM in conjunction with STEM. We not only measured the details of the atomic arrangement and strain gradient by STEM but also visualized a flexoelectric field by detecting depolarized electric fields using the DPC-STEM technique. We determined the intrinsic flexopiezoelectric coefficient to be ~30 J C⁻¹ through the direct observation of the lateral piezoresponse (~6 pm V⁻¹) and strain gradient (~10⁶ m⁻¹) at the ferroelastic domain walls in epitaxial WO₃ films. Based on the lateral flexopiezoelectricity at the twin walls and mechanical switching, and considering the strain-gradients, we suggest that the shear strain-gradients at the twin walls of a soft material induce flexoelectric polarizations and sufficiently detectable piezoresponses at the twin walls.

## Methods

**Synthesis of epitaxial thin films**. Epitaxial WO₃ thin films were grown on orthorhombic YAlO₃ substrates with an (110)_YAO surface by pulsed laser deposition. WO₃ layers were grown at 600–800 °C with an oxygen pressure of 70–100 mTorr. After deposition, they were cooled down to room temperature with an oxygen pressure of 500 mTorr. Our films stabilized the monoclinic structural phase γ-WO₃ (space group P2₁/n, $a = 7.306$ Å, $b = 7.540$ Å, $c = 7.692$ Å, $\beta = 90.88°$ in bulk, where each lattice parameter corresponds to twice the pseudocubic lattice parameter, JCPDS card no. 72–0677). A detailed x-ray diffraction analysis was reported in a previous paper[25].

**TEM and STEM**. Two types of TEM samples were prepared in this study: a cross-section view (having [001]_YAO, [1Ī0]_YAO, and [1Ī1]_YAO projections) and a plane-view ([Ī10]_YAO projection). Cross-sectional TEM samples for structural analysis were prepared by focused ion beam milling (JIB-4601F, JEOL) and subsequent low energy Ar⁺ ion milling to remove the damaged surface layer (PIPS, Gatan). Cross-sectional bright-field (BF) and dark-field (DF) TEM imaging were performed with a 200 kV field-emission TEM (JEM-2100F, JEOL) equipped with a spherical aberration corrector (CEOS GmbH). Electron diffraction patterns were acquired separately for each domain using the smallest selected area aperture and compared with simulated ones with SingleCrystal (CrystalMaker software Ltd) (Supplementary Fig. 2).

Unlike the cross-section view TEM samples, the plane-view TEM sample for the atomic scale strain analysis was prepared by back side (substrate side) milling, including mechanical flat polishing to a thickness of <10 µm and Ar⁺ ion milling using 3.3 kV and 8°. The domain structure of the WO₃ thin film is preserved by the interaction between the film and substrate, which is known as the clamping effect. However, as the substrate gets thinner while preparing a plane-view sample, the domain structure gets weaker and is easier to collapse. For this reason, the plane-view TEM sample was prepared with extreme care to maintain the domain structure, using low energy and the angle of the Ar⁺ ion beam (1 kV and 1°), as the sample became thinner (PIPSII, Gatan).

The BF TEM and high-angle annular dark-field (HAADF) STEM images were obtained using a STEM (JEM-ARM200F, JEOL) at 200 kV equipped with a 5th-order probe-corrector (ASCOR, CEOS GmbH)[42–44] at Materials Imaging &

Analysis Center of POSTECH in Republic of Korea. The collection semi-angles of the HAADF detector were adjusted from 68 to 280 mrad to collect scattered electrons in a large angle for clear Z-sensitive images. To measure the strain gradient at atomic scale, a probe size of ~0.8 Å and camera length of 8 cm were used. The obtained HAADF STEM images were band-pass Wiener filtered to reduce the background noise (Filters Pro, HREM research Inc.). The coordinate information of the atomic positions was extracted using PPA (Peak Pairs Analysis, HREM Research Inc.). The strain analysis on an atomic scale was conducted using lab-made scripts based on Python to precisely determine the positions of the atomic columns at sub-pixel resolution. The STEM images were treated to reduce background noise and to extract each atomic column position with sub-pixel accuracy. A denoising autoencoder, a kind of machine learning technique based on fast Fourier transformation, was used to reconstruct the STEM images to avoid the image distortion from conventional filtering method. Each STEM image was sliced into image patches containing a single atom, as training data or input data. The atomic image patches were reconstructed via denoising function and unsupervised training was conducted. These processes minimize the differences between a noisy input image and a reconstructed output image. After sufficient training, reconstructed image patches are formed without noise, assembled again to construct the full STEM images. Finally, all the atomic positions are extracted and calculated to analyze the lattice distortion on an atomic scale using Python based handmade scripts.

**DPC- STEM**. To visualize the electric polarization at the domain walls, we used the DPC-STEM technique. When an electron beam goes through a TEM specimen, the beam can be deflected by electric fields inside the sample[36–38]. To directly observe the internal electric field for the top view and side view DPC-STEM imaging, the TEM specimens prepared for TEM/STEM analysis were used. DPC-STEM imaging was performed at 200 kV aberration corrected STEM (JEM-2100F, JEOL) equipped with a segmented annular all field detector (SAAF) with eight segments. The angular ranges from the optical axis of detector segments 1–4 and 5–8 were 0–24 and 24–48 mrad, respectively. To enhance the DPC sensitivity, the semi angle of the probe forming the condenser lens aperture was set at 24 mrad, and a camera length of 50 cm was used. All DPC-STEM images were taken along the zone axis conditions to prevent electron deflection caused by misorientation. The electron beam was oriented parallel to the $[1\bar{1}0]_{YAO}$, $[001]_{YAO}$, and $[1\bar{1}1]_{YAO}$ for the side view DPC-STEM imaging (Supplementary Figs. 21 and 22) and to $[\bar{1}\bar{1}0]_{YAO}$ for the top view DPC-STEM imaging, respectively. Analysis of the DPC-STEM images was performed online using a direct reconstruction system.

**Angle-resolved lateral PFM**. Lateral PFM images were obtained at room temperature using scanning probe microscopy (Bruker, MultiMode-V with a Nanoscope controller V) using Pt-coated Si conductive tips (MikroMasch, HQ:DPER-XSC11). An ac driving voltage of 2 V at 5 kHz was applied to a conductive tip while the mechanical vibrations of the tip were monitored by a photodiode segmented into four quadrants using the lock-in amplifier technique, to scan the sample surface at a speed of 0.5 μm s$^{-1}$. The spring constant of the tip was ~2.7 N m$^{-1}$ for the lateral PFM imaging, and it was ~42 N m$^{-1}$ for the mechanical switching. The normal force on a film surface for usual PFM scans was a few nN, while the normal force for the mechanical switching was two orders of magnitude higher than the usual PFM scans. All PFM measurements including the mechanical switching were executed in a relative humidity of about 40–50%.

The lateral PFM measures the tip oscillation in the torsional vibration mode according to the converse piezoelectric effect. Because only the component projected perpendicular to the tip axis is measured, multiple scans of an identical area using nonparallel tip directions are required to construct the full vector information of the local piezoresponses. To map the lateral piezoresponse vectors on the WO$_3$ film surface, four PFM images were taken for the same region with 45°, 135°, 225°, and 315° orientations between the tip and $[001]_{YAO}$. The reason for this is that the lateral displacement is detected as a tip torsional mode, which is normal to the cantilever axis as a projected component. The signals appear or fade away depending on the tip orientations, so that the piezoresponse vector is oriented parallel to the domain walls. The lateral piezoresponse vector at each spatial position can be quantitatively determined by integrating all the scan data and performing trigonometric curve fitting for the measured signals as a function of the tip-orientation angle. By trigonal fitting the in-plane real signals as a function of the orientation angle, position-sensitive angle-resolved PFM images were gained, as shown in Fig. 2e. The lateral displacement at the wall was calculated using a previously obtained factor for our system[4,32,33] (Supplementary Note 3).

**Finite-element modeling**. The shear strain on the WO$_3$ surface induced by the frictional force of an AFM tip was simulated using finite-element analysis. We used the ANSYS Mechanical APDL 14.5 software (from ANSYS Inc.). For the simulation, an isotropic and elastic 2D layer of WO$_3$ was introduced with an elastic constant of ~10$^{11}$ Pa and Poisson's ratio of ~0.25 (refs. [27,28]). Constraints on the four fixed boundaries were exerted, and a frictional force of 40 nN on the tip area of the WO$_3$ surface was presumed. The size of the 2D layer was 120 nm × 120 nm with 18 × 18 elements, and the diameter of the tip contact area was assumed to be ~15 nm.

## Data availability

The data that support the findings of this study are available from the corresponding authors on reasonable request.

## Code availability

The analysis and simulation codes used for this study are available from the corresponding authors on reasonable request.

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

## Acknowledgements
This work was supported by a National Research Foundation (NRF) of Korea Grant funded by the Korean Government through the Creative Research Center for Lattice Defectronics (Grant No. NRF-2017R1A3B1023686) and the Center for Quantum Coherence in Condensed Matter (2016R1A5A1008184). The work of K.S. was supported by a NRF grant funded by the Korean government (MSIT) (Grants No. NRF-2018R1A2B6008258) and the Fundamental Research Program of the Korea Institute of Materials Science (KIMS) (Grant No. PNK6410). S.-Y.C. acknowledges the support of the Global Frontier Hybrid Interface Materials of the National Research Foundation of Korea (NRF) funded by the Ministry of Science and ICT (2013M3A6B1078872) and POSTECH-Samsung Electronics Industry-Academia Cooperative Research Center.

## Author contributions
S.Y. and C.-H.Y. conceived and designed the project. S.Y., J.S., C.-S.W. and C.-H.Y. prepared the samples. S.Y. and K.C. measured and analyzed the PFM images. K.C. performed phase field simulation. K.S., S.-Y.H., G.-Y.K. and S.-Y.C. carried out the TEM and DPC-STEM measurements. S.Y., K.S., S.-Y.C. and C.-H.Y. led the manuscript preparation with contributions from all authors.

## Competing interests
The authors declare no competing interests.
