## [Peer Review File · Nature Communications]

Editorial Note: Parts of this Peer Review File have been redacted as indicated to maintain the confidentiality of unpublished data. Parts of this peer review file have been redacted as indicated to remove third-party material where no permission to publish could be obtained.

Reviewers' comments:

Reviewer #1 (Remarks to the Author):

This paper "Flexopiezoelectricity at ferroelastic domain walls" reports the enhanced piezoelectric response at the wide domain wall. Recently the flexo effects have attracted a lot of attentions. Also many efforts have been made to explore the properties of domain walls as they have potential applications in nanodevices. In this paper, they used various PFM, TEM methods and simulations to characterize the domain walls in WO₃ and conclude that the enhancement come from the flex effect due to the large strain gradient across the domain wall. Overall, I think these findings are interesting. However, at the current stage, I find that the conclusions are not well supported by the provided data.

1. One of my main concerns is that I notice the strain gradient is one of the most important results. It was obtained from the TEM image. It's well known that sample preparation for the TEM is the destructive and strain relaxation is inevitable. In this sense, I think the authors need to provide other characterization to further support this conclusion.
2. The TEM image is only a 2D projection of 3D structure. The measured domain wall thickness and strain gradient assumed that the domain walls are ideally perpendicular to substrate. However, I cannot find data to support this point. In fact, the DPC data in Fig. S4b shows the domain walls are not ideally along any crystallographic planes.
3. In page 7, for the plane-view sample, the substrate is polished away. The strain conditions should be very different than the thin film geometry. The authors already noticed that "The domain structure of the WO₃ thin film is sustained by the interaction with the film and substrate, which is known as 15 the clamping effect". The authors should discuss these issues.
4. Page7, the polarization is estimated to be $\sim 13 \mu\text{C}/\text{cm}^2$. I also worry about there are too many parameters including flexovoltage from the literature to extract the value. Can the authors directly determine the polarization by measure the atom displacement? or other methods.
5. The authors did some semi-quantitative analysis of the properties. The details are included in the supplementary materials. This is very nice indeed. However, I am again a bit worry about the accuracy. I would suggest them to include phase filed simulations, which should be very helpful to verify these results or even extract more interesting conclusions.
6. In fig.1a, the domain walls are not straight in PFM, different than that in the TEM image in Fig.s2. In addition, in the inset in Fig.3a, the inclined stripes correspond to the domain wall between A/B, while the horizontal and vertical lines should be fine DWs. It seems that the distances between A/B and find domain wall are comparable, which is very different than the PFM data in Fig. 1a. Again, does this mean strain relaxation in the TEM samples?
7. The details of recording conditions for the dark field images in Fig.1 should be included.
8. In fig.3, the strain gradient is calculated across the domain wall, while the polarization in Fig.2 is along the domain wall. Is this inconsistent? or the authors did not address this clearly?
9. Fig.3, how did the authors do the strain analysis? It should be clarified.
10. In Fig.4, regarding the DPC measurement, as far as I know, either strain, electric field, or specimen tilt (maybe more factors) can generate the contrast. How did the authors exclude these effects?
11. Still about the DPC measurement, can the author quantify the strength of fields? If so, it will be very useful to determine the magnitude of polarization.
12. Did the authors measure the E_y along other direction? is it consistent?
13. Fig.5, it is very interesting to see the mechanical probe can switch the domain pattern. It looks the domain wall contrast is enhanced and domain wall density is reduced. Did the authors characterize it by TEM after switching?
14. In Supplementary Fig.1, the simulated electron diffraction pattern does not match the experimental one very well. For example, the reflections at -300, -500, -520 are seen in c while not in b. Difference also exists between e and f. Please clarify these differences.

15. In supplementary Fig.4, the scale bar is missing.

16. The title should be more specific by including WO₃. I think the case in this study is not the common one, because many publications show that the thickness of ferroelastic domain walls is only one-two unit cells (for example in BiFeO₃, PbTiO₃).

Reviewer #2 (Remarks to the Author):

By combinations of PFM, TEM and STEM techniques, this work reports special piezoelectricity and polarization at the ferroelastic domain walls of a dielectric oxide-WO₃, and the so-called flexopiezoelectricity, induced by the intrinsic strain gradient at ferroelastic domain walls. Indeed, strain gradient may introduce polarization in oxides through flexoelectric effect, which is a hot topic of oxide research areas. This effect is dominant at nano scale since small dimension crystals could accommodate large strains and strain gradient, which is important for new device concepts, such as piezoelectric effects with no piezoelectric materials.

This is an interesting work which shows that considerable piezoelectric response could be observed at ferroelastic domain walls exhibiting strain gradient. However, the experiments, and related deductions, do not strongly support the PFM observations.

I have two major concerns: 1) the strain states, and the relationship between strain states and the observed piezoelectric effect. In this manuscript the lateral PFM signals were obtained in the as-grown films; however, the strain gradients were measured in a TEM sample, where the substrate clamping was removed. The domain structure may be still the same, but the strain states in a freestanding, very thin WO₃ membrane should be relaxed, and should be very different from that of the as grown epitaxial films. Thus, the relationship of the measured strain gradients and piezoelectric response observed here, the present deduction in this manuscript should be reconsidered.

2) the DPC TEM measurements. I note that there are plenty of domain walls in this WO₃ film. It should be very very careful for a DPC measurement to eliminate possible diffraction effects. It seems that the plenty of domain walls (including the fine domain wall) might contribute strong diffraction contrast to the DPC results, especially for the cross-section samples, whose domain wall inclined ~45 degree to the beam direction. Thus, the DPC results should be reconsidered as well.

In summary, if the strain gradient and electric field obtained here are flawed, the observed PFM signals may originate from other factors, such as point defects, or other polar defects at domain walls.

Here I also have some more discussions with the authors:

1: The width of domain walls. How do the authors get this result of ~20nm? I note that the domain wall of the macro type is not a straight line in Fig 1a. Steps and kinks may evolve in a ferroelastic walls. How could the author exclude this effect? This effect may also contribute contrast to the DPC measurement, which further complex the signals acquired in the DPC experiments. In particular, I find that the strain maps in Supplementary Fig. 3 might be originated from a different method other than the PPA. Moreover, there are obvious differences of the domain wall widths in Supplementary Fig. 3b, where the middle wall is much narrower than the left two walls. This indicates the evaluations of domain wall width, and thus the strain gradient, and related deductions, should be reconsidered.

2: Based on HAADF-STEM imaging (for instance, Ultramicroscopy 160, 57 (2016); Science 348, 547 (2015); Nature Commun. 8:15994 (2017)), strains and strain gradients could be measured accurately

by geometric phase analysis, especially for large scale strain and strain gradient analysis (like Supplementary Fig. 3 here). The fine analysis of strain, strain gradient, and domain wall width may be facilitated by geometric phase analysis.

3: Strain gradient also exists at the fine walls. Why the so-called flexopiezoelectricity was observed only at macro walls?

4: Avoid using the words like "astonishing" and "unprecedented".

Reviewer #3 (Remarks to the Author):

The authors report on the enhancement of the piezoelectric response at the ferroelastic domain walls of WO₃ thin films. Using DPC-TEM, the strain gradient induced polarization are visualized on the atomic scale. They also demonstrate the domain wall manipulation by scanning the surface with an AFM tip generating surficial strain gradient.

The contents of this paper are technically thorough, and the conclusions drawn are satisfactory. However, the motivation of the work in the broader context is unclear and the implication of the work needs to be spelled out more specifically. For example, are the results found specific to WO₃ strained to YAlO₃ (110)? How can the domain wall thickness can be controlled? To me, I couldn't appreciate where the achievement of this work could lead to. I would like to ask the authors to address the above point as well as the following technical questions before further consideration:

1. It is well-known that WO₃ accommodates oxygen vacancies, especially in thin films. Could the authors comment on the role of oxygen vacancies and ferroelastic properties in their samples? For example, do they have evidence of reduced ferroelasticity when samples are grown in reducing conditions?

2. In Fig. 5, they show the PFM images before/after scanning. I am slightly concerned whether humidity during the tip-scan has any effect. Can the authors comment on this?

Reviewer #4 (Remarks to the Author):

The paper is an important contribution to domain boundary engineering and should be published after significant modifications. The experimental results are important and deserve publications. My comments refer mainly to some misunderstandings and missing previous publications on the same topic.

From the outset, the authors have misunderstood the relevance of WO₃. WO₃ is not a 'canonical' ferroelastic material (like SrTiO₃ and many others). There are two reasons. First, WO₃ contains 13 crystallographic phases (or more) so that the determination of a 'canonical' ferroelastic order parameter is impossible without taking into account the coupling between these order parameters. Second, the traditions are mainly driven by electronic effects (First-principles reinvestigation of bulk WO₃ Hamdi, Hanen; Salje, Ekhard K. H.; Ghosez, Philippe; et al. PHYSICAL REVIEW B Volume: 94 Issue: 24 Article Number: 245124 Published: DEC 19 2016).

The motivation for PFM studies of WO₃ (and the reason why this paper is really so important) is that WO₃ shows superconductivity of the domain walls (Sheet superconductivity in twin walls: experimental evidence of WO_{3-x} Aird, A; Salje, EKH JOURNAL OF PHYSICS-CONDENSED MATTER

Volume: 10 Issue: 22 Pages: L377-L380 Published: JUN 8 1998) Therefore, many previous attempts were made to deposit WO₃ thin film and generate twin boundaries (Ferroelastic twin structures in epitaxial WO₃ thin films Yun, Shinhee; Woo, Chang-Su; Kim, Gi-Yeop; et al. APPLIED PHYSICS LETTERS Volume: 107 Issue: 25 Article Number: 252904 2015, Characterization of WO₃ thin films prepared by picosecond laser deposition for gas sensing Preiss, Elisabeth M.; Krauss, Andreas; Kekkonen, Ville; et al. SENSORS AND ACTUATORS B-CHEMICAL Volume: 248 Pages: 153-159 2017).

The same study as in the submitted manuscript was done by in a free standing WO₃ sample in: Nanoscale properties of thin twin walls and surface layers in piezoelectric WO_{3-x} Kim, Yunseok; Alexe, Marin; Salje, Ekhard K. H. APPLIED PHYSICS LETTERS Volume: 96 Issue: 3 Article Number: 032904 Published: JAN 18 2010. This paper was a milestone and showed the increased conductivity of the twin walls and the reduced piezoelectricity. This is in contradiction with the submitted results. This problem needs to be solved. A large part of the paper needs to be dedicated to this issue.

The focus on the flexoelectricity is fine but somewhat trivial. All findings and explanations have been published before so that it is embarrassing to read the paragraph before equ.1 which is largely incorrect but also has been published correctly many times before. The reference to the Stengel papers is sufficient. In case of WO₃, the domain structure is much more complex to be covered in this way. In fact, the intersection problems and the increasing complexity was explained in: Flexoelectricity and the polarity of complex ferroelastic twin patterns Salje, Ekhard K. H.; Li, Suzhi; Stengel, Massimiliano; et al. PHYSICAL REVIEW B Volume: 94 Issue: 2 Article Number: 024114 Published: JUL 25 2016.

The reader is slightly put off by wrong and unfortunate expressions like 'monoclinic collision'. These are not dynamic processes. Therms like 'show up' is not in this context related to any science, etc. I recommend that some native English speaker corrects the paper if resubmitted.

The list of critique is rather long and severe. However, the experimental findings are highly relevant. At this point it is unclear whether they are correct or not. The contradiction with the Kim et al paper is a key issue. So far there is little evidence that superconducting twin boundaries exist in deposited thin films. This manuscript contains aspects which come close to a reliable observation. Unfortunately, it contains many incorrect statements and an insufficient appreciation of the relevant prior results.

RESPONSE LETTER

Review#1-General :

Reviewer #1 (Remarks to the Author):

This paper “Flexopiezoelectricity at ferroelastic domain walls” reports the enhanced piezoelectric response at the wide domain wall. Recently the flexo effects have attracted a lot of attentions. Also many efforts have been made to explore the properties of domain walls as them have potential applications in nanodevices. In this paper, they used various PFM, TEM methods and simulations to characterize the domain walls in WO₃ and conclude that the enhancement come from the flex effect due to the large strain gradient across the domain wall. Overall, I think these findings are interesting. However, at the current stage, I find that the conclusions are not well supported by the provided data.

Response :

We would like to thank the reviewer for their constructive comments on our paper which have helped us significantly improve the manuscript in the new form. In this regard, we have done our best to address or incorporate all of the reviewer’s suggestions, as described in detail below. We hope that the reviewer is convinced by our all efforts.

Review#1-1 :

1. One of my main concerns is that I notice the strain gradient is one of the most important results. It was obtained from the TEM image. It’s well known that sample preparation for the TEM is the destructive and strain relaxation is inevitable. In this sense, I think the authors need to provide other characterization to further support this conclusion.

Response :

We agree with the reviewer’s opinion. The mechanical boundary condition in a TEM specimen is different from a film on substrate and thus the ferroelastic twin structure could be sensitively influenced by the mechanical condition, and such modification could be more severe in the geometry of a planar view. So, it is essential to handle the samples carefully with a minimized

dose of electron beams near the edge of a gradually thinned area. Nonetheless, we often found the occurrence of rapid strain relaxation and we were disappointingly forced to repeat the fabrication/measurement process. Indeed, we performed an extremely delicate work with considerable effort. At the very first time, it was also challenging to get TEM images even for a cross-sectional view. The twin structure at the initial stage in 70 nm-thick WO_3 film was easily relaxed by illumination of a weak electron beam with a flux of 14 pA/cm^2 (see the Supplementary Video R1). It implies that the twin structures are more likely wrinkled with a strong strain gradient rather than are involved with the surface reconstruction and/or the significant defect accumulation.

This technical issue was resolved by increasing the film thickness from 70 nm to hundreds of nm, enabling us to explore even atomic-scale high-resolution images with the stronger electron energy and dose. The typical size of domains increases with a film thickness, according to the scaling of a Kittel's-law type, and a larger energy is required in a thicker sample to change the overall domain pattern through strain relaxation, thereby having more endurance against perturbations. Furthermore, the details of the TEM measurement to minimize the strain relaxation issue will be explained later in this response letter.

As asked by the reviewer, we also introduce an experimental result measured in a non-destructive way to support the existence of strain gradients at ferroelastic domain walls. We carried out x-ray diffraction to examine the diffusive nature of $\text{WO}_3 (001)_{\text{pc}}$ peak. Figure R1 is a quasi-HK reciprocal space map obtained by multiple ω -rocking scans at different azimuthal angles. From this map, we were able to identify the mosaic rotations of monoclinic unit cells by observing the $(001)_{\text{pc}}$ peaks on the H or K axes with a deviation from the center by the mosaic rotation angle. There are four kinds of monoclinic unit cell rotations in ferroelastic fine domains, which are labeled as A_1 , A_2 , B_1 , and B_2 . The shape of a peak has anisotropy in peak broadness, arising from the narrow width of a ferroelastic domain and the elongation along the other in-plane axis.

In addition, the peaks are diffusively connected to each other creating a weak and broad structure on each of quadrants. Since A_1 (or A_2) and B_1 (or B_2) domains subject to mosaic rotations around H or K-axis meet at macro-domain walls (green and blue lines in Fig. R2a), the domain wall regions are expected to be tilt toward $\langle \text{HH} \rangle$ directions. Provided that an intermediate strain state with a sizable area exists at the domain walls, a measurable peak is

expected to emerge at in-plane azimuthal directions toward one of 45° , 135° , 225° , and 315° , which are represented as M_1 , M_2 , M_3 and M_4 , respectively (Figure R2b).

Fig. R1 | X-ray Diffraction of ferroelastic domains and walls. **a**, a semi-HK reciprocal space map at WO_3 (001), and its cross-sections for the ~ 420 -nm-thick film. A_1 , A_2 , B_1 , and B_2 represent the diffraction peaks of ferroelastic domains (fine-domains) with the different mosaic rotation angles. M_1 , M_2 , M_3 , and M_4 indicate the intermediate phases at ferroelastic domain walls (macro-domain walls) which have a mosaic rotation angle of 45° , 135° , 225° , or 315° , respectively. Green, magenta, and red lines are cross-sections to obtain the FWHMs of A_1 and M_4 peaks. **b**, Deconvolution of the cross-sections. FWHMs for the deconvoluted peaks are indicated with a unit of nanometers.

Indeed, we were able to observe the diffusive peaks. Such a diffusive feature between M peaks and A & B domain peaks suggests that there are structural gradients in the vicinity of macro-domain walls. To estimate the width of ferroelastic domain walls, the full-width-at-half-maximum (FWHM) of M_4 peak was analyzed through deconvolution of peaks for a 45° magenta cross-sectional profile in Fig. R1a (see the details in Fig. R1b). Interestingly, the calculated wall width is ~ 26.5 nm ($= \sim 3.71 \text{ \AA} / 0.014$), which is consistent to the measured value (~ 20 nm) by the PFM as well as TEM experiments. Thus, this non-destructive characterization enhances the credibility of the claim for gradual structural changes across the macro-domain walls.

Fig. R2 | Schematics of WO_3 ferroelastic domains. **a**, Top-view schematic with three different ferroelastic domain walls. The thin (or thick) black line represents the fine-domain wall (or super-macro-domain wall). Green and blue lines indicate macro-domain walls. The intermediate phases of M_1 and M_3 (or M_2 and M_4) in Figure R1 compose blue (or green) macro-domain walls. **b**, 3D schematic with monoclinic unit-cells. Grey (or pink) unit cell represents A (or B) macro-domain. At the boundaries between A and B domains (green and blue lines), the intermediate phases with 45° , 135° , 225° , and 315° mosaic rotations are expected. The number of monoclinic unit cells in a fine-domain depends on the film thickness.

Review#1-2 :

2. The TEM image is only a 2D projection of 3D structure. The measured domain wall thickness and strain gradient assumed that the domain walls are ideally perpendicular to substrate. However, I cannot find data to support this point. In fact, the DPC data in Fig. S4b shows the domain walls are not ideally along any crystallographic planes.

Response :

Most of domain walls are perpendicular to the substrate as clearly seen in the low-magnification TEM image of Fig. 1b. However, domain walls are locally not perpendicular to substrate and slightly inclined (as indicated by red arrows in Fig. R3), resulting in a deviation of domain width from the standard value. It is unclear whether the special region is produced

through a partial relaxation in a TEM specimen or it is the original structure. But it occupies a minor portion.

Fig. R3 | Low-magnification TEM image of the 300 nm-thick $\text{WO}_3/\text{YAlO}_3$ sample fabricated by FIB. There are locally slightly inclined domains as indicated by the red arrows, however, the domain walls are mostly almost perpendicular to the YAO substrate.

The domain wall width is directly measured with STEM images. In STEM images, the macro-domain walls appear with a brighter contrast due to the electron scattering at domain walls by irregular atomic arrays (atomic dechanneling). Fig. R4 is the example figures for estimating the domain wall width. We took an area for making the average contrast profile across a macro-domain wall (See Fig. R4a) and we measured the FWHM of the Gaussian fit function (See Fig. R4b). We similarly examined the FWHMs of 18 domain walls, and an average domain wall thickness was found to be 17.85 nm with a standard deviation of 3.03 nm.

Fig. R4 | Example of direct measurement of the domain wall width in STEM images. a, Two macro domain walls are indicated by yellow arrows. The box with the solid red line is the selected domain wall area to measure the domain wall width. The intensity profile of this area is shown in **b. b,** The measured domain wall width is approximately 16.71 nm.

Review#1-3 :

3. In page 7, for the plane-view sample, the substrate is polished away. The strain conditions should be very different than the thin film geometry. The authors already noticed that “The domain structure of the WO₃ thin film is sustained by the interaction with the film and substrate, which is known as the clamping effect”. The authors should discuss these issues.

Response :

Some of domain structures of the WO₃ thin film are sustained by the interaction with the film and substrate, known as the clamping effect. Therefore, as the plane-view sample gets thinner, the domain structure gets relaxed due to the weaker clamping effect. For this reason, the plane-view TEM sample was prepared with the extreme care to maintain the domain structure using the very low energy. The mild condition of Ar⁺ ion beam (1 kV and 1°) is applied to the 10 μm-thin foil sample. After the ion-milling process, the thinnest region of the sample is pierced by the beam, forming the thickness variation along the perimeter of the hole (See Fig. R5 a and b). In Fig. R5b, Area “A” is the thinnest area just around the hole. Area “C”, which is too thick to be transparent to the electron beam, exhibits the residual YAlO₃ substrate above the WO₃ film. The STEM image of these regions is displayed in Fig. R5c. Area “C” is too thick to see any contrast by the transmitted electron beam. On the other hand, the area “A” is too thin to see any domain structure because the area “A” is so away from the substrate that the clamping effect between the film and the substrate is largely annihilated. Interestingly, the domain structure at the area “B” is still visible, even though the region no longer has a substrate to support after polishing. As a result, the area “B” is an optimal area which is thin enough to be capable of the STEM analysis on the domain structure. However, this area is non-uniformly distributed throughout the sample depending on how the remained YAlO₃ substrate looks, so it is challenging to create an ideal area fitting to TEM/STEM imaging. FFT patterns of the areas “A” and “B” also imply the same results (Fig. R5d).

Fig. R5 | Plane-view sample preparation and residual clamping effect of free-standing film region. **a**, Schematic of a thin film. The out-of-plane direction is $[1\bar{1}0]_{\text{YAO}}$. **b**, Schematic of plane-view sample processing. The substrate is mechanically polished away and then the ion beam further thins the sample until the sample is pierced. Film thickness varies depending on the distance from the hole. **c**, Low magnification ABF STEM image around the hole. The darker area implies the thicker area. “A”, “B”, and “C” in **b** and **c** are correspondent each other. In “B”, the stripes indicate that the domain structure remains. **d**, FFT patterns at the regions of “A” and “B”. The single bright spot at the center in “A” represents no periodic pattern meaning that the domain structures are disappeared, while the FFT pattern from “B” is indicative of the periodic patterns, formed by the domain structures.

Review#1-4 :

4. Page7, the polarization is estimated to be $\sim 13 \mu\text{C}/\text{cm}^2$. I also worry about there are too many parameters including flexovoltage from the literature to extract the value. Can the authors directly determine the polarization by measure the atom displacement? or other methods.

Response :

We agree with the reviewer's opinion. Instead of estimating the polarization from the strain analysis by considering flexoelectric coefficients, the measurement of atomic displacements more intuitively provides the polarization information. However, there is a trade-off between the strain measurement and atomic displacement measurement. The thin plane-view TEM sample can provide all atomic positional information of W and O atoms but such a thin TEM sample is vulnerable to the relaxation of the twin structure. So we can't measure the flexoelectric polarization with such a thin plane-view sample. On the other hand, we can perform the strain analysis using the relatively thick plane-view TEM sample (>100 nm), which makes inevitably impossible to measure the oxygen atomic positions and thus it is impossible to measure the accurate the polarization from W and O atomic positions. That's why we had to estimate the polarization by considering the strain by using a thick sample and the flexoelectric coefficient.

Review#1-5 :

5. The authors did some semi-quantitative analysis of the properties. The details are included in the supplementary materials. This is very nice indeed. However, I am again a bit worry about the accuracy. I would suggest them to include phase field simulations, which should be very helpful to verify these results or even extract more interesting conclusions.

Response :

As suggested by the reviewer, we performed an intensive phase field simulation of the WO_3 system based on our Landau free energy density. Through this revision process, we had a great opportunity to significantly improve the manuscript through the theoretical visualization in addition to the semi-quantitative estimates. The details of the model and how to determine the model parameters are summarized in a separate section of the Supplementary Information. Our system was reduced into a two-dimensional model assuming the layer was identical along the thickness direction, and the periodic boundary condition was applied along the in-plane directions. The spatial sampling rate was set to be 5 nm per pixel and the pixel grid size is 80-by-80, so the resultant dimension of the model space is $400 \times 400 \times 5 \text{ nm}^3$.

We can successfully stabilize the hierarchical domain structure consisting of fine-domains and macro-domains using the Metropolis algorithm as shown in Figs. R6-8. The visualization of

the strains, strain gradients, and the electric polarizations over the domains and across the domain walls provides useful insight into the twin structure. The macro-domain walls have a wide width of ~ 30 nm. The emergence of the flexoelectric polarizations at the macro-domain walls is also reproduced. The magnitude of the in-plane polarizations is $\sim 8 \mu\text{C}/\text{cm}^2$ (Fig. R8b), which is of the same order of magnitude as the analytically estimated value.

Fig. R6 | Simulation result of the strain and polarization fields. a, Spatial distribution of strains. **b,** Spatial distribution of ferroelectric polarization. The x and y axes are parallel to the edges of the plot and the z -axis is out of the plane.

From the phase field simulation, we can not only verify the experimental observations, but also extract the additional features we didn't pay attention to before. A significant value of P_z is observed at fine domain walls due to the large shear strain gradient across the thin walls. In reality, two neighboring fine-domains face each other with a sharp interface. It brings us an open question whether the Landau theory is still valid for describing discontinuities or abrupt

jumps in the order parameter field at the atomic scale where the continuum approximation is failed.

Another interesting point is that the hierarchical twin structure can be stabilized without the substrate constraint. The structure itself is in a meta-stable state. We can interpret that the misfit strain from substrate is used for selection of the structure among various possible candidates. Proper matching of the lattice parameters between the structure and the substrate realize the hierarchical twin structure with minimal deformation. So, we can understand that some regions of a TEM specimen for the planar view still endure the relaxation.

Fig. R7 | Strain gradient maps and cross-sectional profiles. a, Strain gradient maps. The z -axis derivatives are all zero so the corresponding images are omitted. The x and y axes are parallel to the edges of the plot and the z -axis is out of the plane. **b, Line profiles of two chosen strain gradients,** which show gradual and stiff changes of strains at the macro-domain walls (left) and fine-domain walls (right), respectively.

Fig. R8 | Line profiles of strain and polarization in the $x'y'z'$ Cartesian coordinate. a, Line profiles of strain and strain gradient along x' -axis. **b,** Line profiles of ferroelectric polarization components which are parallel (y') or perpendicular (x') to domain walls within the surface.

Review#1-6 :

6. In fig.1a, the domain walls are not straight in PFM, different than that in the TEM image in Fig.s2. In addition, in the inset in Fig.3a, the inclined stripes correspond to the domain wall between A/B, while the horizontal and vertical lines should be fine DWs. It seems that the distances between A/B and fine domain wall are comparable, which is very different than the PFM data in Fig. 1a. Again, does this mean strain relaxation in the TEM samples?

Response :

We are sorry that the surface topographic image in Fig. 1a looks confusing. The macro-domain walls are straight along the white dashed lines in the figure. The crooked lines near the domain walls are the step edges with a single unit-cell height change, which indicates the film is grown in the step-flow growth mode. If one carefully looks at the domain walls guided by the dotted lines and ignore the crooked lines, it will be realized that all domain walls are actually straight. The step-terrace structure due to the substrate miscut angle is superimposed with the surface height change attributed to the hierarchical ferroelastic domain structure. The fact that the detailed surface feature at the atomic scale can be clearly seen in a 610-nm-thick film guarantees the surface flatness and excellent crystallinity of the sample. We appreciate the reviewer's comment and leave the information of step-terrace structure in the caption of Fig. 1a in the revised manuscript.

The relative ratios of the length scales (macro-domain, fine-domain, domain wall width) can be different depending on film thickness. More quantitative investigations have been made and are presented in the Fig. R21.

It is a great intuitive comment to point out the stripes in the inset of Fig. 3a. We thank the reviewer for letting us to pay more attention to this strange contrast. The sparse vertical or horizontal line contrast in the domain A and B in Fig. 3a most likely originates from the fine domain walls. (It gives a chance to assess the sharp interface between fine domain walls in contrast to the broad strain gradient region of the macro-domain walls.) However, the distance between the two lines doesn't represent the exact width of the fine domains, since the fine domains are considered to be partially relaxed. The fine domains are formed as a result of small mosaic rotations sharing the *a* or *b* axis, so they are easily relaxed into a single merged domain.

[Redacted]

Figure R9 | TEM study for a 70-nm-thick WO₃ film. (APL 107, 252904 (2015)) (a) The WBDF TEM image of the as-prepared sample showing a highly strained fine domain structure. (b) The fine domain structure easily relaxed by illumination of a weak electron beam with a flux of 14 pA/cm². The scale bars indicate 20 nm.

Such weakness of the fine domains was already shown in our preceding work (APL 107, 252904 (2015)), showing that the sizes of macro-domains and fine domains depend on the film thickness and accordingly the strain field. In case of the 70-nm-thick WO₃ film, the fine domains are fragile; and the fine domains disappear after exposure to a weak electron beam, as shown in Fig. R9. Due to its structural fragility, the ~300-nm-thick WO₃ film was fabricated in the current study. From the plane-view results of Figures R5 and R10, the macro-domains can be clearly identified and also the fine-domains in Fig. R10b show that the width of fine-

domains is ~ 10 nm, which is much smaller than that of the macro-domains. However, some of fine domains are relaxed and disappear. At the left and right-side regions of Figure R10a, the stripes originated from the fine domain walls are largely diminished.

Figure R10 | Macro-domain walls and fine-domain walls from the plane-view TEM observation. **a**, Macro-domains and fine domains are observed. **b**, A magnification view of the area indicated by the yellow dashed box in **a**. The fine-domain walls with a width of ~ 10 nm periodically appear.

In brief, the stripes in Fig. 3a represents the fine domain walls but much more fine domain walls might be relaxed and become invisible. The main purpose to perform the plane-view analysis is to unveil the macro domain walls and thus we didn't describe such contrasts in detail. In the revised manuscript, we added the explanation about the contrast in the Supplementary Information.

Review#1-7 :

7. The details of recording conditions for the dark field images in Fig.1 should be included.

Response :

To clearly understand ferroelastic twin structure, we obtained dark-field TEM images using various g -beams (Fig. R11). The dark-field (DF) TEM image shown in Fig. 1b was taken with reflection $g=00-2$. Following the reviewer's recommendation, we revised the Fig. 1b and added

the g -beam condition of DF TEM image in the figure caption.

Fig. R11 | BF-TEM image taken along zone axis $[001]_{\text{YAO}}$ and DF-TEM images under various g -beam conditions. Twin domains of WO_3 film are rotated by $\pm 45^\circ$ relative to $[001]_{\text{YAO}}$ zone axis. Two beam DF-TEM images with $g=[002]$, $g=[00-2]$ show the domain structure more clearly.

Review#1-8 :

8. In fig.3, the strain gradient is calculated across the domain wall, while the polarization in Fig.2 is along the domain wall. Is this inconsistent? or the authors did not address this clearly?

Response :

The gradient of the shear strain component ($\epsilon_{x'y'}$) across the domain wall (x') induces the polarization along the domain wall (y'). The flexoelectric coefficient $\mu_{y'x'x'y'}$ is responsible for the coupling, i.e., $P_{y'}^f = 2\mu_{y'x'x'y'} \frac{\partial \epsilon_{x'y'}}{\partial x'}$. The relationship can be more intuitively understood by considering the case of a rod that is bent by 90° , where the flexoelectric polarization is induced along the symmetric axis within the plane defined by the two non-parallel segments of the bending rod. More detailed information is provided in the Supplementary Text 7.

Review#1-9 :

9. Fig.3, how did the authors do the strain analysis? It should be clarified.

Response :

We analyzed the stain distribution by measuring atomic position in the atomically resolved STEM HAADF image. The atomic positions were extracted by using the developed scripts by Python for a better. The detail was updated in the “Methods” of revised manuscript, as follows.

“The STEM images were treated to reduce background noise and to extract each atomic column position with sub-pixel accuracy. A denoising autoencoder (DAE), a kind of machine learning technique based on fast Fourier transformation, was used to reconstruct the STEM images to avoid the image distortion from conventional filtering method. Each STEM image was sliced into image patches containing a single atom, as training data or input data. The atomic image patches were reconstructed via denoising function and unsupervised training was conducted. These processes minimize the differences between a noisy input image and a reconstructed output image. After sufficient training, reconstructed image patches are formed without noise, assembled again to construct the full STEM images. Finally, all the atomic positions are extracted and calculated to analyze the lattice distortion on an atomic scale using Python based hand-made scripts.”

Review#1-10 :

10. In Fig.4, regarding the DPC measurement, as far as I know, either strain, electric field, or specimen tilt (maybe more factors) can generate the contrast. How did the authors exclude these effects?

Response :

As the reviewer pointed out, DPC STEM is very sensitive to strain, electric field, thickness variation, and diffraction condition. Firstly, to reduce the experimental error caused by local thickness variations of a TEM sample, TEM samples was fabricated by FIB. The FIBed sample is relatively thick so that strain can maintain and the film thickness variation can be minimized.

The schematic illustration of the 8 segmented detectors used for this study is shown in Fig. R12a, wherein the yellow disk indicates a bright-field beam. As the reviewer pointed out, tilting a TEM specimen is crucial in DPC STEM imaging to avoid dynamical diffraction effects. We performed the DPC STEM analysis by tilting a sample far away from zone-axis to minimize effect of diffraction contrast. However, the tilting range was quite limited because our WO₃ film structure is very complicating, in which twin domains are rotated by $\pm 45^\circ$ relative to [1-10]_{YAO} or [001]_{YAO} zone axes.

Therefore, to avoid the diffraction contrast, we obtained DPC images under two different conditions; at the zone-axis condition and off the zone-axis conditions. Figure R12b shows the cross-sectional low-magnification BF-TEM image of WO₃/YAlO₃ sample at the [1-10]_{YAO} zone-axis condition, where the bright and dark diffraction contrasts correspond to A and B domains, respectively. The intensity of DPC STEM image arises from the beam deflection, which is proportional to the magnitude of the electric field [Lohr, M., Schregle, R., Jetter, M., Wächter, C., Wunderer, T., Scholz, F., Zweck, J. Differential phase contrast 2.0—Opening new “fields” for an established technique. *Ultramicroscopy* **117**, 7–14 (2012); Naoya Shibata, N., Scott D. Findlay, S. D., Sasaki, H., Matsumoto, T., Sawada, H., Kohno, Y., Otomo, S., Minato, R., Ikuhara, Y. Imaging of built-in electric field at a p-n junction by scanning transmission electron microscopy. *Scientific Reports* **5**, 10040 (2015)]. If the detector segments 5-8 of layer 2 are used for electric field mapping and their orientations are aligned as shown in Fig. R12a, the total beam deflection along the x and y direction can be calculated by using the simple relationship as follows.

$$d_x = (5+8) - (6+7)$$

$$d_y = (7+8) - (5+6)$$

We present the simultaneously-obtained DPC STEM images from detector segments 5-8 and calculated DPC STEM images corresponding to d_x and d_y (Figs. R12c-e).

Fig. R12 | DPC-STEM images simultaneously observed by the segmented detector. a, Schematic illustration of the segmented detector used for this study. **b,** BF-TEM image and the corresponding electron diffraction pattern taken along the $[1-10]_{\text{YAO}}$ zone-axis. **c,** DPC STEM images formed by detector segments 5-8 schematically illustrated in **a**. **d, e,** DPC STEM images obtained by calculation according to the orientation relation between sample and detector.

The $(5+8)-(6+7)$ DPC STEM image clearly shows contrast variation across the domain B, while the $(7+8)-(5+6)$ image shows no significant image contrast within the WO_3 film. It means that there are relatively strong electric fields along the in-plane direction. To visualize accurate electric field components (E_x and E_y), we measured the angle between the x direction of segmented detector and the beam deflection direction, and then calibrated results were shown in Fig. R13. Unlike the diffraction contrast shown in the BF-TEM image of Fig. R13a, contrast inversion was observed within B domains in the $E_{[001]_{\text{YAO}}}$ electric-field map (Fig. R13c).

In order to check the effect of diffraction condition, we carried out the DPC experiments at different diffraction conditions. Figure R14 shows the DPC-STEM result taken with slightly tilted away from the zone-axis. Although the contrast in BF-TEM image was changed by diffraction condition, the electric field map along the x direction is almost the same as the result in Fig. R13. This clarifies that the contrast inversion is not caused by diffraction contrast in WO_3 film.

Fig. R13 | Projected electric field vector map using DPC-STEM at the zone axis condition. **a**, BF-TEM image taken along the $[1-10]_{\text{YAO}}$ zone-axis. **b**, Projected electric field vector map. **c**, **d**, E_x and E_y maps obtained from the segmented detectors. Electric field maps along x and y direction were calibrated considering the angle between the x direction of segmented detector and the beam deflection direction.

Fig. R14 | Projected electric field vector map obtained off the zone-axis. a, BF-TEM image and the corresponding diffraction patterns of domain A and B. Sample is slightly tilted away from the $[1-10]_{YAO}$ zone-axis. **b**, **c**, E_x and E_y maps obtained from the segmented detectors. **d**, Projected electric-field vector map. Electric field map was calibrated considering the angle between the x direction of segmented detector and the beam deflection direction.

Beyond the results shown in Figs. R12-14, it is difficult to obtain the intuitive DPC-STEM result because the twin domain walls of WO₃ were inclined 45° with respect to the viewing directions and therefore the domain wall are not sharp, mixed up with two domains. In addition, the intensity signal in the DPC data can be attributed to strain distribution within the WO₃ film. We additionally prepared the TEM sample so that the twin domain walls are sharp, right oriented along a viewing direction. We observed the strain distribution across the twin domain wall in WO₃ by atomic STEM imaging (as presented in Fig. 3 of main text). However, there were no significant contrasts that could arise from strains in the DPC-STEM images under our DPC-STEM analysis condition (Fig. R10). Although the strain and diffraction contrasts may be included in our result, our experimental results so far demonstrate that the interesting signals in the DPC-STEM data are mainly due to the electric fields within WO₃ film rather than the other effects. These data have been added in the revised supplementary information.

Fig. R15 | Electric fields map by DPC-STEM for a diagonal cut. **a**, Schematic view of an experimental geometry. **b**, DPC-STEM images taken for a diagonal cut surface with $[\bar{1}\bar{1}\bar{1}]_{\text{YAO}}$ zone axis. The twin walls of WO_3 film are oriented in a direction parallel to the viewing direction. **b**, Projected electric field maps along the horizontal and vertical directions. **c**, Constructed electric field vector map.

Review#1-11 :

11. Still about the DPC measurement, can the author quantify the strength of fields? If so, it will be very useful to determine the magnitude of polarization.

Response :

At the moment, our DPC-STEM data were obtained by using a SAAF detector with 8 segments (Fig. R12); and thereby we understand the directional information of the field but the quantification is not available. Quantitative measurement of electromagnetic fields is available within the weak phase object approximation (WPOA) by using the phase contrast transfer function (PCTF) [Dekkers, N. H., de Lang, H. Differential phase contrast in a STEM. *Optik* **41**, 452-456 (1974); Rose, H. Nonstandard imaging methods in electron microscopy. *Ultramicroscopy* **2**, 251-267 (1977)]. However, it is also very challenging though. Recently, by defining the center of mass (CoM) of electron distribution in the transmitted bright field disk via a pixelated detector, the quantification of fields has been facilitated under the phase object approximation [Müller, K. Krause, F. F., Béch e, A., Schowalter, M., Galioit, V., L offler, S., Verbeeck, J., Zweck, J., Schattschneider, P., Rosenauer, A. Atomic electric fields revealed by a quantum mechanical approach to electron picodiffraction. *Nature Communications* **5**, 5653 (2014)].

However, we have improved the DPC-STEM data by comparing the relative signal strengths obtained from different cuts of a sample with respect to the noise level in vacuum. (Please see the Supplementary Fig. 21 d and c and the related caption.)

Review#1-12 :

12. Did the authors measure the E_y along other direction? is it consistent?

Response :

We also performed DPC imaging along the other directions to confirm the DPC result (Fig. S4 and Fig. R16). The DPC image presented in the main manuscript corresponds to the side view 1, and the other in-plane orientation was also investigated to get the electric field component along the y direction (side view 2). Although the observed region is complicated by the narrow domains and deformation as shown in the figure, the electric field profile is consistent with the original DPC result taken through the side view 1. These data have been added in the revised supplementary information.

Fig. R16 | Electric fields at the polar domain walls constructed using the DPC STEM. a, Schematic of the zone-axes for the DPC STEM imaging. **b,** The cross sectional BF-TEM image taken along the $[001]$ direction of YAO substrate. **c,** E_y (left), E_z (center) and vector map (right)

for selected area indicated by red rectangle in **b**. The horizontal components (E_y) strongly contribute to the DPC STEM image.

Review#1-13 :

13. Fig.5, it is very interesting to see the mechanical probe can switch the domain pattern. It looks the domain wall contrast is enhanced and domain wall density is reduced. Did the authors characterize it by TEM after switching?

Response :

The ferroelastic domain walls in WO_3 films show a hierarchical structure. Stripes of fine-domains define a macro-domain. A bundle of stripe macro-domains defines a single super-macro-domain. In contrast to the well-defined straight stripe fine- or macro-domain walls, the super-macro-domain walls appear to be not well ordered. Provided a hierarchical ordering continued, we could get a fractal-like self-similar structure (stripe super-macro-, super-super-macro-, ... domains and walls). In the as-grown region, scattered super-macro-domains are observed and their irregularity can induce mechanical frustration in some regions. In the situation of constraints, we can understand the macro-domain width is not entirely uniform. However, the poled area was subjected to a frictional force that aligns the macro-domain walls, leading to a well-defined single super-macro-domain. We can get more uniform stripes inside the poled box.

Regarding the intensity of in-plane PFM signals, we cannot identify a meaningful difference between the poled areas and the as-grown area. Probably, the shaded boxes overlaid on the poling regions seem to be misleading (Fig. R17). In the revised manuscript, we have removed the shaded color boxes. When the boxes are removed (Fig. R17b), the written areas do not look like having higher contrast than the as-grown area. The mechanical writing process makes the ferroelastic domain configuration be changed into an ordered form with more even and wider domain width than before, which may be more stable. The more irregular domain structure in the as-grown area is almost frozen by the quenching of a configuration during the cooling process. We previously tried to switch the domain structure using in-situ AFM experiments, however we failed to get the meaningful results due to its extreme difficulty. We were not able

to further characterize the mechanical switched structure by TEM because of the limited time of revision. Taking into account no significant change in PFM signals, we expect that there may be only changes in domain orientation and width.

Fig. R17 | In-plane PFM images after the mechanical switching. a, Image with shaded boxes representing switched areas. **b**, Image without shaded boxes. Scale bars represent 1 μm .

Review#1-14 :

14. In Supplementary Fig.1, the simulated electron diffraction pattern does not match the experimental one very well. For example, the reflections at -300, -500, -520 are seen in c while not in b. Difference also exists between e and f. Please clarify these differences.

Response :

The twin domain walls in WO_3 were inclined 45° with respect to the in-plane $[1\bar{1}0]$ and $[001]$ directions of YAlO_3 substrate, and the specimen for TEM analysis was prepared thicker than the typical TEM specimen to prevent the domain relaxation. Therefore, domains in the cross-sectional TEM can overlap with different domains along the viewing direction. In addition, since the width of the domains is not constant as shown in Fig. R18, it is highly likely that the diffraction patterns will be overlapped. Although we used the smallest SA aperture about 150 nm in diameter to acquire the diffraction pattern in each domain, the projected diffraction pattern obtained in one domain may slightly overlap with the diffraction from the neighboring domains. Because of this, some forbidden reflections are observed in the experimental

diffraction patterns. However, the strong reflections indicated by the red, orange, and yellow colored circles in the experimental diffraction patterns of Figs. R18 c and d result from the proper crystallographic symmetry, and they are consistent with the simulated ones. We provide a cautionary note of the possible overlap of neighboring domains through the zone-axis in the Supplementary text related to the Fig. S1. We sincerely appreciate the reviewer's careful reading and constructive comments.

Fig. R18 | Simulation of electron diffraction patterns of WO_3 . **a**, The cross sectional BF-TEM image taken along the $[1\bar{1}0]$ of YAO substrate. **b**, TEM image showing size of selected area (SA) aperture. The smallest SA aperture was used for domain structure analysis and its diameter is about 150 nm. **c**, Experimental (left) and simulated electron diffraction pattern (center) taken along the $[010]$ direction of WO_3 monoclinic structure (right). **d**, Experimental (left) and calculated electron diffraction pattern (center) taken along the $[\bar{1}00]$ of WO_3

monoclinic structure (right). The simulation of electron diffraction was constructed by CrystalMaker software using WO_3 monoclinic structure.

Review#1-15 :

15. In supplementary Fig.4, the scale bar is missing.

Response :

We would like to thank the reviewer for very careful comment. We added the scale bar in Fig. S4 (currently Figs. S21 & S22 of the revised manuscript).

Review#1-16 :

16. The title should be more specific by including WO_3 . I think the case in this study is not the common one, because many publications show that the thickness of ferroelastic domain walls is only one-two unit cells (for example in BiFeO_3 , PbTiO_3).

Response :

As asked by the reviewer, we have modified the title be more specific with including WO_3 .

Review#2-General :

Reviewer #2 (Remarks to the Author):

By combinations of PFM, TEM and STEM techniques, this work reports special piezoelectricity and polarization at the ferroelastic domain walls of a dielectric oxide-WO₃, and the so-called flexopiezoelectricity, induced by the intrinsic strain gradient at ferroelastic domain walls. Indeed, strain gradient may introduce polarization in oxides through flexoelectric effect, which is a hot topic of oxide research areas. This effect is dominant at nano scale since small dimension crystals could accommodate large strains and strain gradient, which is important for new device concepts, such as piezoelectric effects with no piezoelectric materials.

This is an interesting work which shows that considerable piezoelectric response could be observed at ferroelastic domain walls exhibiting strain gradient. However, the experiments, and related deductions, do not strongly support the PFM observations.

Response :

We thank the reviewer for finding important points in our work and giving us thoughtful comments on the manuscript. Our detailed responses to each of the comments are described below.

Review#2-1 :

I have two major concerns: 1) the strain states, and the relationship between strain states and the observed piezoelectric effect. In this manuscript the lateral PFM signals were obtained in the as-grown films; however, the strain gradients were measured in a TEM sample, where the substrate clamping was removed. The domain structure may be still the same, but the strain states in a freestanding, very thin WO₃ membrane should be relaxed, and should be very different from that of the as grown epitaxial films. Thus, the relationship of the measured strain gradients and piezoelectric response observed here, the present deduction in this manuscript should be reconsidered.

Response :

It is a valid question. The alternate emergence of a - and b -domains is helpful in minimizing the misfit strain energy because the in-plane lattice parameters of NdAlO₃ substrates are between the a - and b -axis lattice parameters of WO₃, through a commensurate matching over a supercell. In addition, each of a - or b -axis oriented macro-domains contain small fine-domains a^+ and a^- or b^+ and b^- , which are subject to mosaic rotations so that the c -axis points to the normal of substrate. The hierarchical twin structure seems to be reinforced by the substrate clamping. But, we realize that a more correct expression is that the twin structure excellently harmonizes with the substrate. It is a matter of whether the twin structure itself is sustainable or it is forced by external stress. In other words, provided that the hierarchical twin structure is a meta-stable point in the free energy landscape, we can understand the sustainability of the structure in TEM specimens regardless of no substrate. In fact, the terminology of “clamping” gives a misleading impression that the structure is forced to appear by an external stress and the twin structure is originally located at an unstable point in the free energy landscape without the substrate. In reality, the substrate clamping plays a crucial role in protecting the meta-stable structure from perturbations by increasing the thermodynamic barrier. In the revised manuscript, we correct the description.

Of course, the uniform single domain structure is the most stable in the absence of mechanical constraints, so strong perturbation relieves the twin structure to a more merged configuration step by step. The fact that one can find a specific example that shows that the twin structure remains in an unrelaxed form even in the region without a substrate is a strong evidence of the meta-stable configuration. Moreover, the phase field simulation without consideration of the substrate clamping also stabilize the hierarchical twin structure (see the corresponding section in the Supplementary Information), theoretically supporting the meta-stable hypothesis. Furthermore, the lattice parameters of WO₃ films don't change over a broad range of film thickness. Even if we have grown a lot of WO₃ films varying the film thickness from a few 10 nm to a few 100 nm and up to ~1 micron thickness, they all have the same lattice parameters. The 1-micron-thick film also shows a nice step-terrace structure on the surface morphology, indicating the film is still atomically flat. It means an exact matching between the substrate and film. We are not sure it is just a fortune in substrate selection and/or it is attributed to the fact that WO₃ has a variety of competing tilt structures.

In the follows, we describe our TEM measurement process for the geometry of planar view in a technical point of view in detail and provide a representative image with unrelaxed fine-domains. These parts alongside Figs. R19 and R20 are partially duplicated with the previous answers for the Review #1-3 and Review #1-6.

Fig. R19 | Plane-view sample preparation and residual clamping effect of free-standing film region. **a**, Schematic of a thin film. The out-of-plane direction is $[1\bar{1}0]_{\text{YAO}}$. **b**, Schematic of plane-view sample processing. The substrate is mechanically polished away and then the ion beam further thins the sample until the sample is pierced. Film thickness varies depending on the distance from the hole. **c**, Low magnification ADF STEM image around the hole. The darker area implies the thicker area. “A”, “B”, and “C” in **b** and **c** are correspondent each other. In “B”, the stripes indicate that the domain structure remains. **d**, FFT patterns at the regions of “A” and “B”. The single bright spot at the center in “A” represents no periodic pattern meaning that the domain structures are disappeared, while the FFT pattern from “B” is indicative of the periodic patterns, formed by the domain structures.

As the substrate gets thinner for preparing a plane-view sample, the domain structure gets weaker and easier to collapse. For this reason, the plane-view TEM sample was prepared with extreme care to maintain the domain structure using the very low energy. The mild condition of Ar^+ ion beam (1 kV and 1°) is applied to the 10 μm -thin foil sample. After the ion-milling process, the thinnest region of the sample is pierced by the beam, forming the thickness variation along the perimeter of the hole (See Fig. R19 a and b). In Fig. R19b, the area “A” is

the thinnest area just around the hole. Area “C”, which is too thick to be transparent to the electron beam, exhibits the residual YAlO_3 substrate above the WO_3 film. The STEM image of these regions is displayed in Fig. R19c. Area “C” is too thick to see any contrast by the transmitted electron beam. On the other hand, the area “A” is too thin to see any domain structure because the area “A” is so away from the substrate that the clamping effect between the film and the substrate is largely annihilated. Interestingly, the domain structure at the area “B” is still visible, even though the region no longer has a substrate to support after polishing. The area “B” is an optimal area which is thin enough to be capable of the STEM analysis on the domain structure. However, this area is distributed non-uniformly throughout the sample, so it is challenging to create an ideal area fitting to TEM/STEM imaging. FFT patterns of the areas “A” and “B” also imply the same results (Fig. R19d).

From the plane-view results of Figures R20, the macro-domains can be clearly identified and the fine domains in Fig. R20b shows that the width of fine domains is ~ 10 nm, which is much smaller than that of the macro-domains. However, some of fine domains are relaxed and disappear. At the left and right-side regions of Figure R20, the stripes originated from the fine domain walls are largely diminished.

Figure R20 | Macro-domain walls and fine-domain walls from the plane-view TEM observation. **a**, Macro-domains and fine domains are all survived. **b**, The magnified area indicated by the yellow, dotted box. The fine domain width is ~ 10 nm and the fine domain walls are periodically appeared.

Review#2-2 :

2) the DPC TEM measurements. I note that there are plenty of domain walls in this WO₃ film. It should be very very careful for a DPC measurement to eliminate possible diffraction effects. It seems that the plenty of domain walls (including the fine domain wall) might contribute strong diffraction contrast to the DPC results, especially for the cross-section samples, whose domain wall inclined ~45 degree to the beam direction. Thus, the DPC results should be reconsidered as well.

Response :

As the reviewer pointed out, the diffraction contrast sensitively influences DPC imaging. Accordingly, the DPC images were recorded in conditions far away from zone-axis to minimize effect of diffraction contrast.

In our study, we found out that the magnitude of the lateral piezoresponse is significantly large at the domain walls and their directions are parallel to the domain walls from angle-resolved PFM analysis (Fig. 2). To directly image the electric field expected to exist through the piezoresponse and strain gradient, we carried out the DPC experiments for the plane-view and cross-sectional samples. Unfortunately, for the cross-section sample where domain walls are parallel to the incident electron beam direction and domains is not overlapping, DPC signal was not detected because the electric field is parallel to the incident electron beam direction. Therefore, we carried out the additional DPC experiments of cross-sectional TEM sample where twin domains are rotated by $\pm 45^\circ$ relative to $[1-10]_{\text{YAO}}$ or $[001]_{\text{YAO}}$ zone axes. We carefully checked the effect of diffraction contrast at various diffraction conditions and then obtained DPC images, as shown in Figs. R12-R15. Moreover, for intuitive analysis, DPC experiments were performed on a plane-view TEM sample. As expected, the electric fields arising from the interfacial flexoelectricity are clearly detected at the macro-domain walls in the plane-view sample (Fig. 4b).

We provided various vector maps obtained by DPC experiments and discussed this point on page 8 of the original submitted manuscript, which read: Because the domain walls are inclined to the zone axis by 45° , the expected electric fields at the walls are projected onto the cross-sectional plane.

Extensive DPC analyses have been made for many different directional cuts regardless of painful repetitions. Our efforts and reasoning to eliminate possible diffraction effects were described in detail in the Review#1-10. All DPC results and the relevant procedures for DPC-STEM analysis are added in the revised supplementary information.

Review#2-3 :

In summary, if the strain gradient and electric field obtained here are flawed, the observed PFM signals may originate from other factors, such as point defects, or other polar defects at domain walls.

Response :

As the reviewer mentioned, although we detected the strain gradient based on the position of W ions in TEM measurements, we should consider point defects such as oxygen vacancies. If oxygen vacancies were the main effect of the observed piezoelectric response at macro-domain walls, oxygen vacancies accumulated at the walls, as donors, should have induced high electronic conduction compared to domains.

[Redacted]

It is also worthwhile mentioning that oxygen vacancies expand the volume of sample in oxides in general. The local confinement of oxygen vacancies at the domain walls (aligned along y' -axis) likely induce a strain gradient of $\frac{\partial \varepsilon_{x'x'}}{\partial x'}$ along the perpendicular in-plane axis (x' -axis) with opposite signs on either side of the domain wall. It leads to the flexoelectric polarizations along the perpendicular directions ($+x'$ and $-x'$ directions). It is in contrast to our observation that the flexoelectric polarizations are parallel to the domain walls (along $+y'$ or $-y'$ direction, depending on the shear strain gradient $\frac{\partial \varepsilon_{x'y'}}{\partial x}$). Based on the reasoning symmetry-wise, it is difficult to explain the flexoelectric polarization and the corresponding piezoelectric response

using the oxygen vacancies hypothesis. On these grounds, we can exclude the existence of a significant effect of oxygen vacancies at the macro-domains.

[Redacted]

Figure R21 | Electronic conduction on the surface of a WO₃ film. a, Surface morphology measured by atomic force microscopy. The white dashed lines represent the macro-domain walls. The crooked lines correspond to step edges with a single unit-cell height. [Redacted]

Review#2-4 :

Here I also have some more discussions with the authors:

1: The width of domain walls. How do the authors get this result of ~20nm? I note that the domain wall of the macro type is not a straight line in Fig 1a. Steps and kinks may evolve in a ferroelastic walls. How could the author exclude this effect? This effect may also contribute contrast to the DPC measurement, which further complex the signals acquired in the DPC experiments. In particular, I find that the strain maps in Supplementary Fig. 3 might be originated from a different method other than the PPA. Moreover, there are obvious differences of the domain wall widths in Supplementary Fig. 3b, where the middle wall is much narrower than the left two walls. This indicates the evaluations of domain wall width, and thus the strain gradient, and related deductions, should be reconsidered.

Response :

The surface topographic image of a ~610-nm-thick WO₃ thin film (Fig. 1a, the same as Fig. R21a) shows a hierarchical twin structure on the background of a step-terrace structure with single-unit-cell (less than 4 Å) steps that indicates the film was grown in a step-flow growth mode. It is remarkable that such large-thickness film still has an atomically flat surface to the extent that the peaks and troughs of fine domains due to the mosaic rotations ($\pm 0.82^\circ$) are clearly seen. The AFM sensitivity along the normal is quite good enough to see atomic scale height changes, so we can say that the topographic modulation (seemingly rough) is tiny and negligible compared with the lateral length scale and film thickness.

In Fig. 1a, somewhat irregular step edges look close to the macro-domain walls (straight along the dashed lines which are defined along the meeting points of the *a*-domains and *b*-domains) and the step edges seem to influence on the macro-domain walls nearby. But, we believe the effect was not significant and the similar alignment happened by accident. The step edge directions and widths are determined by uncontrolled misaligned cutting of the substrate ($\sim 0.1^\circ$), so we can find many counter examples where the correlations between macro-domain walls and step edges are weak. It is worthwhile mentioning that the widths of macro-domains are 200-500 nm at the film thickness of 610 nm.

The narrow widths (~ 20 nm) of macro-domain walls is hard to be observed in the topographic image in contrast to the wider widths of the domains, because the macro-domain wall region does not necessarily cause a gradual bending of the fine-domain walls. According to the phase field simulation (not shown here), the deviation of the fine-domain walls from the straight $\langle 100 \rangle$ line can be recognized within a narrow width around the center of the macro-domain wall. The width is as wide as the fine-domain wall width. Since the width of fine-domain walls is at the atomic scale (a few unit-cells), it is difficult to identify a topographic variation in a ~ 20 -nm-wide macro-domain wall region, except for the central line where *a*-domain and *b*-domain meets. Whereas, careful in-plane PFM measurements and the TEM analysis sensitive to strain gradient/defects enable us to clearly identify the widths of macro-domain walls.

As film thickness (*t*) is decreased, the macro-domain width (*w*) is also scaled down, following a power law of $w \sim t^{0.6}$ [S. Yun *et al.*, APL 107, 252904 (2015)]. On the other hand, the macro-

domain *wall width* (a few 10 nm) doesn't sensitively depend on the film thickness. So, the relative surface coverage of the macro-domain wall region increases with decreasing film thickness. Quantitative investigations have been already made and they are discussed in Fig. R25 and the related text below. Regarding the variation in macro-domain widths in a 300-nm-thick film (Supplementary Fig. 3b (in the original version)), the region has somewhat smaller widths of macro-domains. It is natural that the macro-domain widths are not exactly equal, unless the variation is not logarithmically different.

The extraordinarily large width of the ferroelastic domain wall is due to the elastic softness of WO₃. The empty space in A-site is intuitively thought to be a microscopic origin of the large endurance of lattice deformation and strain gradient. The domain wall width can be estimated based on the Landau theory*. The smaller the Ginzburg (gradient) energy, the wider the domain wall width. In this case, the order parameter is the shear strain ($\varepsilon_{x'y'}$). The coefficient of the quadratic square term of a shear strain gradient $\frac{\partial \varepsilon_{x'y'}}{\partial x'}$ is responsible for the domain wall width.

*Ginzburg energy and domain wall width

Let there be a system described by the order parameter ϕ , and the free energy density of the system is given by $F = -\frac{1}{2}\phi^2 + \frac{1}{4}\phi^4 + G \left(\frac{d\phi}{dx}\right)^2$. The values of the order parameter that give the minimum energy are $\phi = \pm 1$. Now, consider a boundary condition that $\phi|_{x=-\infty} = -1$, and $\phi|_{x=+\infty} = +1$. The differential equation for the order parameter field that minimizes the total free energy is

$$\frac{\delta F}{\delta \phi} = \frac{\partial F}{\partial \phi} - \frac{d}{dx} \left(\frac{\partial}{\partial \left(\frac{d\phi}{dx}\right)} F \right) = 0$$

$$-\phi + \phi^3 - 2G \frac{d^2 \phi}{dx^2} = 0,$$

and its solution that represents the domain wall is $\phi(x) = \tanh \frac{x}{\sqrt{4G}}$. It means the domain wall width is proportional to \sqrt{G} .

Review#2-5 :

2: Based on HAADF-STEM imaging (for instance, Ultramicroscopy 160, 57 (2016); Science 348, 547 (2015); Nature Commun. 8:15994 (2017)), strains and strain gradients could be

measured accurately by geometric phase analysis, especially for large scale strain and strain gradient analysis (like Supplementary Fig. 3 here). The fine analysis of strain, strain gradient, and domain wall width may be facilitated by geometric phase analysis.

Response :

As the reviewer mentioned, we can visualize the strain distribution by Geometric Phase Analysis (GPA) as shown in Fig. R22, and it shows the similar results with Fig. 3b. However, in the Fig. R22b, the results obtained via GPA cannot reveal a delicate strain change across the domain wall. This difference is due to the difference in the way of calculating strain distributions between GPA and atomic scale analysis. GPA calculates strain distribution by comparing FFT throughout the whole images. At the domain wall, the FFT images of two domains are overlapped, so the information of the gradual strain changes across the domain wall must be missing in the highly magnified image.

Fig. R22 | Strain analysis of atomic scale image with Geometric Phase Analysis. a, High resolution STEM images of the domain wall. The white dotted line is along the domain wall. **b**, Strain analysis with GPA for the region of **a**. The black dotted line is along the domain wall and identically located with **a**.

In contrast, we directly extracted the strain distribution from the atomic positions using a high resolution STEM image, which is a useful method for calculating strain and strain gradients on an atomic scale. In particular, the flexoelectricity is proportional to the size of the strain gradient, which is inversely proportional to the domain wall width on an atomic scale of a few Å . Therefore, in order to calculate the size of flexoelectric field more correctly and directly, it is necessary to calculate in units of a few Å . For this reason, we used atomic scale analysis with PPA and Python based scripts.

Review#2-6 :

3: Strain gradient also exists at the fine walls. Why the so-called flexopiezoelectricity was observed only at macro walls?

Response :

As the reviewer mentioned, strain gradient exists at the fine-domain walls as well, and the resultant flexoelectric polarization is along the out-of-plane axis. Surface screening carriers at the fine domain walls significantly suppress the out-of-plane flexopiezoelectric responses (see the answer for Review #4-3). The similar suppression of surface piezoelectricity at twin walls alongside electronic conduction has been reported [Kim Y., *et al.* Applied Physics Letters **96**, 032904 (2010)]. To compare the signals from macro- and fine-domain walls, we executed in-plane and out-of-plane PFM measurements in the same area (Fig. R23). As expected, out-of-plane signals are much smaller than in-plane signals, even though the out-of-plane PFM measurement is much more sensitive. Accordingly, we have mainly focused on flexopiezoelectricity at the macro-domain walls rather than the fine-domain walls.

Figure R23 | In-plane and out-of-plane PFM images for the same area.

Review#2-7 :

4: Avoid using the words like “astonishing” and “unprecedented”.

Response :

As asked by the reviewer, we have removed the words in the revised manuscript.

Review#3-General :

Reviewer #3 (Remarks to the Author):

The authors report on the enhancement of the piezoelectric response at the ferroelastic domain walls of WO₃ thin films. Using DPC-TEM, the strain gradient induced polarization are visualized on the atomic scale. They also demonstrate the domain wall manipulation by scanning the surface with an AFM tip generating surficial strain gradient.

The contents of this paper are technically thorough, and the conclusions drawn are satisfactory.

Response :

We greatly thank the reviewer for this positive evaluation and giving us important comments on the manuscript. In the following, we address the main motivation of our work and answer the questions raised by the reviewer.

Review#3-1 :

However, the motivation of the work in the broader context is unclear and the implication of the work needs to be spelled out more specifically. For example, are the results found specific to WO₃ strained to YAlO₃ (110)? How can the domain wall thickness can be controlled? To me, I couldn't appreciate where the achievement of this work could lead to. I would like to ask the authors to address the above point as well as the following technical questions before further consideration:

Response :

We thank the reviewer for raising the important questions. In the following lines, we would like to deliver the importance of our work.

Since the piezoelectricity was firstly discovered by Pierre Curie and Paul-Jacques Curie in 1880, the piezoelectric phenomena have been merged into our lives with various types of applications such as sonar detector, inkjet printer, microbalances (actuator), ultrafine goniometer, and so on. The piezoelectricity originates from the crystallographic anisotropy, so

called the broken inversion symmetry, and therefore it has been accepted that the piezoelectricity cannot generate in the centrosymmetric materials.

We have been paying attention to a new type of electromechanical response in the centrosymmetric tungsten trioxide due to strain gradient. Despite the existence of crystal inversion symmetry in WO_3 , the detection of an electromechanical response was unusual. Without a new phenomenological term beyond the conventional Landau theory, we cannot describe the electromechanical effect. The observed piezoelectric response was radically distinct from the conventional piezoelectricity and we found that it became enhanced in proportion to the strain gradient. We scrutinized the interrelationship between the strain gradient, piezoelectric response, and the electric polarity using angle-resolved PFM and atomic scale strain analysis in conjunction with the differential phase contrast (DPC) technique, thereby estimating the intrinsic flexopiezoelectric coefficient to be $\sim 30 \text{ J/C [V]}$, which enables the estimation of a flexopiezoelectric response in any given similar class material by multiplying the intrinsic coefficient by elastic compliance, dielectric constant and strain gradient.

Eventually, we propose a new thermodynamic phenomenon so called flexopiezoelectricity; and we have come to realize that this unanticipated observation is not limited to the specific material but a universal phenomenon in nanoscale materials/interfaces subject to strain gradients. The flexopiezoelectric coefficient is represented by a sixth-order (even-ranked) tensor, so the effect can be non-zero regardless of the presence or absence of inversion symmetry. Our findings will have a broad impact on nanoscale materials and device research. For example, corrugated 2D materials have significant transverse strain gradients and, as a result, piezoelectric response should appear according to the knowledge. We expect the piezoelectric response can be generally induced by wrinkling or bending any flexible material.

Furthermore, our finding is helpful in understanding the high-order effects of piezoelectricity or flexoelectricity. We are trying to answer quantitatively and symmetry-wise how piezoelectric coefficients can be modified by strain gradients or how flexoelectric coefficients can be changed by strain. Just as the value of flexovoltage coefficient is universally valid for many perovskite oxides, so is the estimate of the flexopiezoelectric coefficient. On these

grounds, we would like to argue that our study contributes to the fundamentals of electromechanical phenomenology by adding a new missing term.

WO₃ is an example wherein the flexopiezoelectricity manifests itself significantly. The unique features of the compound are that it has a large dielectric constant (~5000) and it is also elastically soft. A large flexopiezoelectric effect at the same intrinsic coefficient can emerge as a result. Many unusual phenomena such as easy ionic penetration/migration, polaronic conduction, superconductivity, and countless polymorphic phases and soft modes, as well as the giant value of dielectric constant and elastic softness in WO₃ and doped systems are based on or closely related to the susceptible nature of such A-site vacant perovskite. In a microscopic point of view, we believe the A-site vacant space acts as a buffer space to endure a large elastic deformation.

Regarding the question of domain wall width, we were also wondering why the macro-domain wall has a wide width while the fine-domain wall appears as a sharp interface as in many other twin walls. The fine-domains form in such a way that the monocline *c*-axis is aligned to the normal of substrate by mosaic rotation. The facing *bc* surfaces of two neighboring fine domains shows good compatibility each other (See Fig. R24a below). So, the sharp interface of two crystals can be energetically stable. On the other hand, the macro-domain wall is at the boundary of two A- and B- domains, of which the in-plane crystal axes are transformed each other by azimuthal rotation of $\pm 90^\circ$ (Fig. R24b). A difference of *a*- and *b*-axis lattice parameters doesn't allow satisfaction of the compatibility relation without a relative deviation of in-plane axis orientations from the substrate axes in domain regions, i.e., violation of the 90° inter-relationship (Fig. R24c). But, since epitaxial coherence is likely to force the orientation matching of the film and substrate in-plane axes, a good solution for compatible interfacial matching cannot be found in the constraint. Therefore, a gradual lattice deformation with a rather wide width occurs in this soft material. If the material is elastically rigid sufficiently, the elastic energy loss is too high to take the deformation. Instead, it is presumed to produce other surface reconstruction alongside defect formation within a relatively narrow interfacial region.

Figure R24 | Compatibility relation. **a**, Schematics of fine-domain unit cells. **b**, Schematic of macro-domain walls. Four-variant monoclinic domains (A_1 , A_2 , B_1 , and B_2) are shown with the color of bright blue, blue, bright yellow, and yellow respectively. Small black arrows represent the directions of monoclinic shear deformation.

As the next question, we need to discuss what determines the width of the domain wall under the assumption that gradual lattice deformation has been chosen as a way to mitigate the interfacial strain in materials with sufficiently low elastic modulus. Domain wall widths are determined by competition between an on-site energy and an inter-site energy (e.g. magnetic anisotropy energy versus magnetic exchange energy in magnetic systems). In this ferroelastic system, the elastic energy such as the Hook's law with a quadratic dependence of strain plays a role of the on-site energy and the gradient energy with a quadratic dependence of strain gradient acts as an inter-site interaction. The smaller the elastic energy compared to the gradient energy, the wider the domain wall width will be.

In the macro-domain wall region, only the shear strain ($\epsilon_{x'y'}$) evolves from a positive value to its opposite negative value across the domain wall along x' -axis. The local strain and strain gradient are relevant to only in-plane axes and seem to be independent of z -axis, i.e., the bending structures within the in-plane are just stacked along the normal of film. So, we guess the macro-domain wall width has little dependence of film thickness. It is quite distinct from the typical macro-domain's width (w_m), which increases with film thickness ($w_m \propto t^{0.6}$). We performed additional PFM characterization on several samples with different film thicknesses (Fig. R25). As compared with the domain width, the macro-domain wall width has almost an identical value ($2 \times \text{FWHM} \sim 30 \text{ nm}$) which is consistent with our expectation. As a result, the surface areal fraction of domain wall regions increases as the film thickness is thinner.

[Redacted]

Figure R25 | Domain-wall width depending on the film thickness. **a**, In-plane PFM images (top) and its cross-sections (bottom) for the four different film thicknesses. White lines in the in-plane PFM images represent the cross-sections. In cross-section graphs, the values of FWHMs are indicated with the unit of nanometers.

[Redacted]

Review#3-2 :

1. It is well-known that WO₃ accommodates oxygen vacancies, especially in thin films. Could the authors comment on the role of oxygen vacancies and ferroelastic properties in their samples? For example, do they have evidence of reduced ferroelasticity when samples are grown in reducing conditions?

Response :

It has been known that oxygen vacancies in a bulk WO_3 tend to induce phase transition from monoclinic to tetragonal phase [Nanoscale properties of thin twin walls and surface layers in piezoelectric WO_{3-x} Kim, Yunseok; Alexe, Marin; Salje, Ekhard K. H. APPLIED PHYSICS LETTERS Volume: 96 Issue: 3 Article Number: 032904]. Based on the Kim's report, we expected that ferroelasticity would be reduced in a WO_{3-x} film. To observe this effect we executed an annealing experiment for 5 hours at 400°C in a vacuum of $\sim 10^{-6}$ Torr. The color of the reduced sample was darker than another as-grown sample (Fig. R26a). However, the ferroelastic structure does not appear to have changed, as measured by atomic force microscopy (Fig. R26 b and c).

Figure R26 | Ferroelastic structure in an oxygen reduced film. a, Picture of annealed film and as-grown film. b,c, Surface morphology (left) with the vertical deflection image (right), which were obtained before (b) and after (c) annealing.

Review#3-3 :

2. In Fig. 5, they show the PFM images before/after scanning. I am slightly concerned whether humidity during the tip-scan has any effect. Can the authors comment on this?

Response :

We totally agree with the opinion that humidity can affect the mechanical writing. It is a common sense that a wet surface is more slippery than the dried one. In other words, water and/or dissociated molecules on the surface most likely reduce the static and kinetic frictional coefficients, thereby hindering the mechanical writing process that relies on the tip-induced frictional force. Tribology on ceramic surfaces alongside examination of humidity-dependent friction have been an extensively studied area of research [Sasaki, S. The effects of the surrounding atmosphere on the friction and wear of alumina, zirconia, silicon-carbide and silicon-nitride. *Wear* **134**, 185–200 (1989); Komvopoulos, K., Li, H., The effect of tribofilm formation and humidity on the friction and wear properties of ceramic materials. *J. Tribol.* **114**, 131–140 (1992); Wäsche, R., Klaffke, D., Troczynski, T. Tribological performance of SiC and TiB₂ against SiC and Al₂O₃ at low sliding speeds. *Wear* **256**, 695–704 (2004); Basu, B., Vitchev, R.G., Vleugels, J., Celis, J.P., van der Biest, O. Influence of humidity on the fretting wear of self-mated tetragonal zirconia ceramics. *Acta Mater.* **48**, 2461–2471 (2000)].

In addition, it has been reported that a smooth (OH⁻) layer forms on various oxide surfaces in environments of high humidity [Perez-Unzueta, A., Beynon, J., Gee, M. Effects of surrounding atmosphere on the wear of sintered alumina. *Wear* **146**, 179–196 (1991); Gee, M.G. The formation of aluminium hydroxide in the sliding wear of alumina. *Wear* **153**, 201–227 (1992); Gates, R.S., Hsu, S.M., Klaus, E.E. Tribochemical mechanism of alumina with water. *Tribol. Trans.* **32**, 357–363 (1989)]. We think that the general tendency could be applied to our experiment either. However, the surface termination (WO₂²⁺-layer or O²⁻-layer) of a film can change the polarity of surface and the resultant molecules on surface can be different. In reality, we observed the interface between film (WO₃) and substrate (YAlO₃) by high-resolution STEM, indicating a WO₂ sub-layer was formed right on the YO layer of the substrate and W atoms occupied only the B-sites of the perovskite keeping the A-site empty (Fig. R27). On the assumption of exclusion of a half-unit-cell growth, the termination of BO₂ at the bottom suggests the AO at the top surface. In this case, positive ionic molecules preferentially reside on the surface to screen the surface charge. This is in a scope of surface chemistry, and much attention is recently paid into the field in the oxide community because many physical/chemical properties of surface/bulk/interface are turned out to depend on the termination and surface treatment.

[Redacted]

Since more specific information on the kinetic friction between the Pt coated tip and the surface of WO_3 film is not allowed at the moment, we refer to humidity effect on the interfacial friction between other metals and oxide surfaces qualitatively. For example, humidity effect on the frictional coefficient between Ni and Al_2O_3 has been known, as shown in Fig. R28. The humidity effect on friction at the metal-oxide interface is much smaller than that at the metal-metal interface [Fukuda, T. & Menz, T. *Micro Mechanical Systems Principles and Technology* (Elsevier science, Elsevier, 1998)]. In the curve of Ni- Al_2O_3 interface, the friction coefficient is reduced by ~33% when the relative humidity increases from ~5% to 70%. We expect a similar effect on the friction between Pt and WO_3 .

Our experiment was executed on 18th September 2018. On that day, the humidity outside was ~67% in Daejeon, South Korea, according to the announcement of the Korean Meteorological Administration [<http://www.kma.go.kr/eng/index.jsp>]. However, the relative humidity in our experimental environment would be about 40~50% because of the operation of an air conditioner in the measurement room. If the humidity effect in our experiment follows the tendency of the case of Ni- Al_2O_3 , we may increase mechanical writing effect by decreasing the humidity. Contrarily, as humidity increases, the mechanical writing effect is reduced but not so much. A higher normal force can compensate the decrease of friction coefficient. To observe the clear humidity dependence, a delicate control of tip contact condition is necessary. For the readers who are interested in the possibility, we leave the information of humidity in the Methods section of the revised manuscript.

[Redacted]

Review#4-General :

Reviewer #4 (Remarks to the Author):

The paper is an important contribution to domain boundary engineering and should be published after significant modifications. The experimental results are important and deserve publications. My comments refer mainly to some misunderstandings and missing previous publications on the same topic.

Response :

We sincerely appreciate the reviewer's positive evaluation on the importance of the manuscript. We reflect the points the reviewer mentioned in the revised manuscript.

Review#4-1 :

From the outset, the authors have misunderstood the relevance of WO₃. WO₃ is not a 'canonical' ferroelastic material (like SrTiO₃ and many others). There are two reasons. First, WO₃ contains 13 crystallographic phases (or more) so that the determination of a 'canonical' ferroelastic order parameter is impossible without taking into account the coupling between these order parameters. Second, the traditions are mainly driven by electronic effects (First-principles reinvestigation of bulk WO₃ Hamdi, Hanen; Salje, Ekhard K. H.; Ghosez, Philippe; et al. PHYSICAL REVIEW B Volume: 94 Issue: 24 Article Number: 245124 Published: DEC 19 2016).

Response :

We appreciate the reviewer's thoroughness in using the exact terminology. WO₃ is one of the most popular ferroelastic materials, but it is not a canonical ferroelastic material that should have a well-defined order parameter with minimizing couplings with other degrees of freedom. WO₃ has a variety of different tilt systems and has many interesting phenomena, such as polaronic conduction and superconductivity, all of them can potentially interact with ferroelastic properties and domain structures. We have replaced the 'canonical' by 'extensively studied' in the revised manuscript. Since the papers mentioned by the reviewer are closely relevant to the study, we cite them in the revised manuscript.

Review#4-2 :

The motivation for PFM studies of WO₃ (and the reason why this paper is really so important) is that WO₃ shows superconductivity of the domain walls (Sheet superconductivity in twin walls: experimental evidence of WO_{3-x} Aird, A; Salje, EKH JOURNAL OF PHYSICS-CONDENSED MATTER Volume: 10 Issue: 22 Pages: L377-L380 Published: JUN 8 1998) Therefore, many previous attempts were made to deposit WO₃ thin film and generate twin boundaries (Ferroelastic twin structures in epitaxial WO₃ thin films Yun, Shinhee; Woo, Chang-Su; Kim, Gi-Yeop; et al. APPLIED PHYSICS LETTERS Volume: 107 Issue: 25 Article Number: 252904 2015, Characterization of WO₃ thin films prepared by picosecond laser deposition for gas sensing Preiss, Elisabeth M.; Krauss, Andreas; Kekkonen, Ville; et al. SENSORS AND ACTUATORS B-CHEMICAL Volume: 248 Pages: 153-159 2017).

Response :

Observation of electronic conduction and possible superconductivity at ferroelastic domain walls is an important motivation for studying the domain walls of WO₃, which we have studied in a parallel way. [Redacted]

[Redacted]

[Redacted]

Considering the strong electron-lattice coupling in the compound (which manifests as a form of bi-polaron hopping), we have a possibility of novel electronic conduction such as superconductivity and spin-orbit coupled phenomena observed in high-Z materials. We need to do more experiments to support this hypothesis. All of these points and references are valuable. We have included them in the revised version.

[Redacted]

Review#4-3 :

The same study as in the submitted manuscript was done by in a free standing WO₃ sample in: Nanoscale properties of thin twin walls and surface layers in piezoelectric WO_{3-x} Kim, Yunseok; Alexe, Marin; Salje, Ekhard K. H. APPLIED PHYSICS LETTERS Volume: 96 Issue: 3 Article Number: 032904 Published: JAN 18 2010. This paper was a milestone and showed the increased conductivity of the twin walls and the reduced piezoelectricity. This is in contradiction with the submitted results. This problem needs to be solved. A large part of the paper needs to be dedicated to this issue.

Response :

Our structural phase is monoclinic (or possibly triclinic), which is different from Kim's sample that is tetragonal, (please refer to our previous paper: Ferroelastic twin structures in epitaxial WO₃ thin films S. Yun, *et al.* APPLIED PHYSICS LETTERS Volume: 107 Issue: 25 Article Number: 252904 2015). Although the base tilt structure is different, we have found electronic conduction at fine-domain walls (Fig. R29) looks similar to the Kim's report. We consider this conduction comes from the flexoelectric effect at fine-domain walls (Fig. R31). Since induced flexoelectric polarization is normal to the film surface, electron carriers are gathered at twin walls to screen the flexoelectric fields. According to the phase field simulation, we expect a sensibly large polarization/piezoelectric response along the out-of-plane direction. But our experimental result doesn't meet the expectation. In contrast to the clear in-plane PFM signal (which is our main focus in the manuscript) at the macro-domain walls, the out-of-plane PFM is largely suppressed (Fig. R23). [Redacted] All these are related to the out-of-plane

polarization/piezoelectric response. Since surface charge density $\sigma = \hat{P} \cdot \hat{n}$ is sensitive to the surface normal (\hat{n}), the out-of-plane geometry is important. But, the macro-domain walls with in-plane polarizations have negligibly small surface charges, enabling us to sense the piezoelectric response. The seemingly different tendency at the two different types of domain walls shows the versatile feature of WO₃.

[Redacted]

Review#4-4 :

The focus on the flexoelectricity is fine but somewhat trivial. All findings and explanations have been published before so that it is embarrassing to read the paragraph before equ.1 which is largely incorrect but also has been published correctly many times before. The reference to the Stengel papers is sufficient. In case of WO₃, the domain structure is much more complex to be covered in this way. In fact, the intersection problems and the increasing complexity was explained in: Flexoelectricity and the polarity of complex ferroelastic twin patterns Salje, Ekhard K. H.; Li, Suzhi; Stengel, Massimiliano; et al. PHYSICAL REVIEW B Volume: 94 Issue: 2 Article Number: 024114 Published: JUL 25 2016.

Response :

As asked by the reviewer, we have modified the introductory part to reflect the more complex nature of domain structures citing the paper of flexoelectricity and the polarity of ferroelastic twin patterns. Also, the paper is highlighted in the Discussion section.

Review#4-5 :

The reader is slightly put off by wrong and unfortunate expressions like 'monoclinic collision'. These are not dynamic processes. Terms like 'show up' is not in this context related to any science, etc. I recommend that some native English speaker corrects the paper if resubmitted.

Response :

We appreciate the reviewer's completeness. As asked by the reviewer, we have removed the words in the revised manuscript. Our revised manuscript was read by a native speaker.

Review#4-6 :

The list of critique is rather long and severe. However, the experimental findings are highly relevant. At this point it is unclear whether they are correct or not. The contradiction with the Kim et al paper is a key issue. So far there is little evidence that superconducting twin boundaries exist in deposited thin films. This manuscript contains aspects which come close to a reliable observation. Unfortunately, it contains many incorrect statements and an insufficient appreciation of the relevant prior results.

Response :

We appreciate the reviewer's constructive comments. As asked by the reviewer, we have improved the manuscript by modifying the introductory part and including the comparison with the result of Kim et al paper. Following the reviewer's recommendation, we cite more papers to contain more appreciation of the relevant prior results. Also, we are very glad to say that our observation at the fine-twin walls is consistent with the Kim et al paper, as described in Review#4-3.

List of changes

1. The changes made during this revision are highlighted in red color in the main manuscript.
2. The Supplementary Information has been greatly improved in an almost new form. It includes most of the new measurement results, analyses, simulations and descriptions/discussions listed in this response letter.
3. The references are also significantly updated to reflect the reviewers' recommendations and improve the manuscript.

REVIEWER COMMENTS

Reviewer #1 (Remarks to the Author):

The manuscript is much improved. Most of my concerns are gone with the additional data. Now I can recommend to publish it. Other minor comments are:

1. The XRD data of domain walls (Figure.R1) seems quite convincing to me to eliminate my concern on destructive sample preparation induced strain relaxation. For the plan view characterization, it may be better to use small thickness sample. In this case, the authors can just leave a relative thick substrate to keep the most of stain impacting on the thin film. Anyway, overall the XRD data is basically convincing to me, at least for the thick sample.
2. The phase field simulation seems consistent, although display goes wrong in the Fig. R6 and Fig.S15. I also notice the domain walls are curved a bit, can the authors explain that?
3. The electric field information of two cross sectional samples with two directions are provided from the DPC measurement. How about the plan view sample? I think it is helpful to include that in the paper as well.

Reviewer #2 (Remarks to the Author):

The authors have addressed most of my concerns. The responses to the questions raised by other three reviewers are also appropriate. This is an interesting work with useful implications for oxide functional materials. I recommend publication of this work in Nature Communications at this stage.

Reviewer #3 (Remarks to the Author):

The authors have addressed my questions and concerns and I recommend the manuscript to be published in the present form.

Reviewer #4 (Remarks to the Author):

In order to save time I have checked the new manuscript with respect to my previous comments. Other pertinent comments by the other referees are left to them to comment upon. I also did not check the supplementary material. I hope that my comments are nevertheless useful

The paper has greatly improved. There are some issues (some are minor):

line 60 replace 'of' by 'between'

line 72 the author need to comment on the symmetry a bit more carefully. The domains in the monoclinic and trigonal phase are different! The distinction between the phases is very simple, best done by Raman spectroscopy. The monoclinic and triclinic phases have very different phonon spectra. Otherwise, x-ray diffraction can also be used. Even in TEM diffraction the difference is visible. Note that the structural changes are NOT simple octahedral tilts but changes of the octahedral shapes. It is important to know which phase is being investigated.

line 94 The statement is unclear. The substrate clamps (otherwise there would be no twin boundaries) and also deforms the layer. This latter effect is mainly due to kinks and steps in the substrate. Both effects are clearly visible.

line 123 do you mean screening? it is not a screen.

I was somewhat alarmed by the response letter: the 3 types of domain walls are not all ferroelastic. The narrow walls are typical W walls and they are ferroelastic. The wider walls are composite walls but still constrained by strain. These walls are fatter which is correctly described in the paper. They can just about be claimed to be ferroelastistic because they confirm macroscopically with the monoclinic symmetry.

The super...(?) wiggly walls are not ferroelastic at all and are not symmetry related. They represent grain boundary type walls between areas in the sample with different local domain wall configurations. They are extremely common in all strained materials, including ferroelastics but they are a consequence of other ferroelastic walls. They are not ferroelastic themselves. Under stress they may even move which is the same effect as the movement of invariant lines in martensites. The authors shouldn't confuse these walls. An 'intrinsic wall thickness' can only be measured for the W walls.

I am not surprised by the 'thickness' of the composite walls. This is not the intrinsic ferroelastic thickness but is due to the extreme roughness of the walls. The models are nice because they show this effect. Other rough walls have the same range of thicknesses. If the authors would have measured on a finer scale (i.e. the atomistic thickness) they would have found the same thickness as the W walls (I would predict).

The comparison with Kim et al basically OK now. In Kim et al. the wall thickness is more intrinsic because of the higher symmetry. What I like about the paper are two aspects. We now know that

1. it is possible to grow twinned WO₃ thin films. This is new.
2. the piezoelectricity is inverted with respect to the Kim et al paper. There the bulk is piezoelectric and the walls are not. Now the bulk is not piezoelectric but the walls are.

I find the fact that the measured walls are thick not terribly exciting because the thickness is basically an artefact of the wall roughness.

Nevertheless, the paper is interesting and can, from my perspective be accepted if a more thorough symmetry analysis and a discussion of the effect of the wall roughness are added.

REVIEWER COMMENTS

Reviewer #1 (Remarks to the Author):

The manuscript is much improved. Most of my concerns are gone with the additional data. Now I can recommend to publish it. Other minor comments are:

Response:

We sincerely thank the referee for the recommendation for publication of our manuscript. The remaining comments are answered as below.

1. The XRD data of domain walls (Figure.R1) seems quite convincing to me to eliminate my concern on destructive sample preparation induced strain relaxation. For the plan view characterization, it may be better to use small thickness sample. In this case, the authors can just leave a relative thick substrate to keep the most of stain impacting on the thin film. Anyway, overall the XRD data is basically convincing to me, at least for the thick sample.

Response:

We are very pleased that the reviewer is satisfied with the XRD data. We are also happy to be able to give the convincing evidence based on the non-destructive method. As the reviewer commented, the area appropriate for the plane-view TEM characterization should be delicately chosen to avoid the strain relaxation.

2. The phase field simulation seems consistent, although display goes wrong in the Fig. R6 and Fig.S15. I also notice the domain walls are curved a bit, can the authors explain that?

Response:

The waviness of the macro-domain walls in the simulation is likely due to random fluctuations at the early stage. Simulating with more iteration steps at higher temperatures is expected to reduce wall fluctuations. In any case, the emergence of the curved shape indicates that the energy cost is not very high. The elastic softness of the material can tolerate wide wall widths with strain gradients as well as fluctuations in the wall centerline. Fluctuations and widths can be correlated, but not a causal effect.

The compatibility relation between monoclinic A1 and B1 ferroelastic domains (or between A2 and B2) predicts a W-type straight ($y=x$) domain wall with the mirror symmetry about the (-101) plane (Fig. R1a). The deviation from the ideal straight line leads to energy costs not only for the macro-domain wall but also for the fine domain walls (Fig. R1b). If the horizontal width of A1 fine-domain is not equal to the vertical width of B1 fine-domain, the junction can deviate from the $y=x$ line (Fig. R1c). More rigorous understanding of the dynamic feature of domain wall fluctuation remains a challenging task at the moment.

Fig. R1, Schematics of domain wall fluctuation. **a**, Ideally, the domain wall between A1 and B1 domains are parallel to the domain wall between A2 and B2 domains. So, the macro-domain wall should be straight. **b**, A bending in the macro-domain wall is accompanied with a curve in the fine domain wall, leading to the increase of the free energy. **c**, Uneven widths of fine domains induce a curve in the macro-domain wall.

3. The electric field information of two cross sectional samples with two directions are provided from the DPC measurement. How about the plan view sample? I think it is helpful to include that in the paper as well.

Response:

For intuitive analysis, we performed the DPC measurement on a plane-view sample. As described in Fig. 4b of the main manuscript, the electric fields are clearly detected at the macro-domain walls in the plane-view sample. Their directions are parallel to the macro-domain walls, and the direction of electric field at the neighboring walls is in the opposite directions.

Reviewer #2 (Remarks to the Author):

The authors have addressed most of my concerns. The responses to the questions raised by other three reviewers are also appropriate. This is an interesting work with useful implications for oxide functional materials. I recommend publication of this work in Nature Communications at this stage.

Response:

We appreciate the reviewer's evaluation that our work is interesting and has useful implications for oxide functional materials.

Reviewer #3 (Remarks to the Author):

The authors have addressed my questions and concerns and I recommend the manuscript to be published in the present form.

Response:

We appreciate the reviewer's recommendation that our manuscript can be published in the present form.

Reviewer #4 (Remarks to the Author):

In order to save time I have checked the new manuscript with respect to my previous comments. Other pertinent comments by the other referees are left to them to comment upon. I also did not check the supplementary material. I hope that my comments are nevertheless useful

Response:

We appreciate the reviewer's precious comments that are useful to make the paper be more balanced and compatible with the historical works of WO_3 .

The paper has greatly improved. There are some issues (some are minor):

Response:

We thank the reviewer for admitting our manuscript was greatly improved. We additionally modified the manuscript to reconcile with the following reviewers' comments.

line 60 replace 'of' by ' between'

Response:

As asked by the reviewer, we replaced it.

line 72 the author need to comment on the symmetry a bit more carefully. The domains in the monoclinic and trigonal phase are different! The distinction between the phases is very simple, best done by Raman spectroscopy. The monoclinic and triclinic phases have very different phonon spectra. Otherwise, x-ray diffraction can also be used. Even in TEM diffraction the difference is visible. Note that the structural changes are NOT simple octahedral tilts but changes of the octahedral shapes. It is important to know which phase is being investigated.

Response:

TEM diffraction patterns were delicately analyzed, as presented in Supplementary Fig. 2. The matching between the measured diffraction spots in the reciprocal space and the expected peaks of the monoclinic phase (space group $P2_1/n$) confirms the crystal structure is monoclinic. The forbidden reflections $\{(h0l): h+l=2n+1, n=\text{integers}\}$ of the monoclinic phase (space group $P2_1/n$) is the basis on which we can disregard the triclinic symmetry (space group $P1\bar{1}$). If it were triclinic, the diffraction spots (e.g. (201), (203), ...) should have had noticeably strong peaks. Regarding the changes of the

octahedron tilts and shapes, we have removed the sentence in the revised manuscript to avoid any confusion. The information on the monoclinic phase is explicitly included in the main text.

line 94 The statement is unclear. The substrate clamps (otherwise there would be no twin boundaries) and also deforms the layer. This latter effect is mainly due to kinks and steps in the substrate. Both effects are clearly visible.

Response:

During the last revision, we have realized that the twin structures can appear without substrate. For example, a TEM specimen fabricated for planar view has a region where the twin structure still survives without relaxation regardless of removing the substrate. Also, we were able to stabilize the twin structure in phase field simulation even if we did not consider substrate clamping. So, we believe the twin structure can maintain as a meta-stable structure (although the structure is not the ground state). The matching with the substrate helps the specific twin structure is chosen among other possible twin configurations. The seemingly interacting feature with the step terrace structure in Fig. 1a is not the general case. Taking a careful look at the topographic image, we notice that the wiggly step edges are overall parallel to the macro-domain walls but the positions are not exactly matched. The wiggly step edges are not a constraint necessary for the twin walls. One clear counter example is shown in Fig. R2.

Fig. R2 | Straightness and even width of macro-domain walls regardless of the step edges in a WO₃ film. Surface topographic image (left) and IP PFM image (right) were measured simultaneously. Black dot lines in the IP PFM image represent the step edges.

line 123 do you mean screening? it is not a screen.

Response:

The sentence “accompanying the screen of the piezoresponses, as in tetragonal WO_{3-x}” is changed to “suppressing the piezoresponses”.

I was somewhat alarmed by the response letter: the 3 types of domain walls are not all ferroelastic. The narrow walls are typical W walls and they are ferroelastic. The wider walls

are composite walls but still constrained by strain. These walls are fatter which is correctly described in the paper. They can just about be claimed to be ferroelastic because they confirm macroscopically with the monoclinic symmetry.

Response:

We totally agree with the reviewer’s insightful opinion. We would like to thank the reviewer for bring us to pay more attention to the definition of ferroelastic domain walls. As pointed out by the reviewer, the super-macro-domain walls cannot be classified as intrinsic W-type ferroelastic domain walls, although the domains/walls can be switched by stress gradients (Fig. 5 in the main manuscript). Among our three types of domain walls, only fine- and macro-domain walls satisfy the requirement of W wall, i.e., the existence of a mirror plane at each wall (see Fig. R3).

We agree that the macro-domain wall is a ferroelastic wall despite it is a composite wall. For example, a macro-domain wall along [110] in-plane direction consists of two intrinsic twin walls; one is a wall between A1 and B1 ferroelastic domains with (-110) mirror plane (symbolized as (A1|B1)₍₋₁₁₀₎, see Fig. R4) and the other is a wall between A2 and B2 ferroelastic domains with (-110) mirror plane (symbolized as (A2|B2)₍₋₁₁₀₎). These two fine domain walls appear alternately along the [110] direction. Different from the curved super-macro-domain walls, these intrinsic fine-domain walls and their composite wall are all straight along [110] directions in terms of in-plane directions. The well-ordered composite structure enables us to examine the width of the intrinsic twin walls.

Fig. R3 | Schematics of WO₃ ferroelastic domains. **a**, Top-view schematic with three different ferroelastic domain walls. The thin (or thick) black line represents the fine-domain wall (or super-macro-domain wall). Green and blue lines indicate macro-domain walls. The intermediate phases of M₁ and M₃ (or M₂ and M₄) in Figure R1 compose blue (or green) macro-domain walls. **b**, 3D schematic with monoclinic unit-cells. Grey (or pink) unit cell represents A (or B) macro-domain. At the boundaries between A and B domains (green and blue lines), the intermediate phases with 45°, 135°, 225°, and 315° mosaic rotations are expected. The number of monoclinic unit cells in a fine-domain depends on the film thickness.

$$\text{Let } \varepsilon_{ik}(A_1) = \begin{bmatrix} -\lambda & 0 & \mu \\ 0 & \lambda & 0 \\ \mu & 0 & \nu \end{bmatrix} \text{ and } \varepsilon_{ik}(B_1) = \begin{bmatrix} \lambda & 0 & 0 \\ 0 & -\lambda & \mu \\ 0 & \mu & \nu \end{bmatrix}$$

B_1 is 90 deg rotation of A_1 along z-axis

Fig. R4 | The possible ferroelastic domain walls of monoclinic A_1 and B_1 domains. The compatibility relation associated with the difference of strain tensors result in two possible domain walls. One is the W-type wall along $[110]$ with the mirror symmetry of (-110) , i.e. $(A_1|B_1)_{(-110)}$. The other is W'-type wall along $[-110]$.

The super...(?) wiggly walls are not ferroelastic at all and are not symmetry related. They represent grain boundary type walls between areas in the sample with different local domain wall configurations. They are extremely common in all strained materials, including ferroelastics but they are a consequence of other ferroelastic walls. They are not ferroelastic themselves. Under stress they may even move which is the same effect as the movement of invariant lines in martensites. The authors shouldn't confuse these walls. An 'intrinsic wall thickness' can only be measure for the W walls.

Response:

We thank the reviewer for the thoughtful comment. As mentioned by the reviewer, the super-macro-domain walls do not have crystallographic mirror planes at all (with wiggles: the superposition of misaligned fine-domain walls and macro-domain walls at the boundary), and they cannot be classified as typical W-type ferroelastic domain walls. We think that the irregular configuration of super-macro-domains is a frozen state (pinned at defects). The configuration as a quenched disorder is far from the most stable ideal configuration. As in glass systems, a configuration can slowly move toward more stable one through thermal annealing and the movement can be accelerated by external perturbation.

I am not surprised by the 'thickness' of the composite walls. This is not the intrinsic ferroelastic thickness but is due to the extreme roughness of the walls. The models are nice because they show this effect. Other rough walls have the same range of thicknesses. If the authors would have measured on a finer scale (i.e. the atomistic thickness) they would have found the same thickness as the W walls (I would predict).

Response:

Although the macro-domain wall is a composite wall, the structure is an ordered form of two intrinsic walls. Our atomic-scale characterization reveals the existence of a large strain gradient in each of intrinsic walls. Intuitively, we can understand that the $(A1/B1)_{(-110)}$ -type fine domain walls are subject to a large strain gradient because the adjacent two domains are rotated by $+ - 90^\circ$ from each other. It is reminiscent of the *a/c* domain walls in PbTiO_3 , where a large strain gradient exists leading to significant flexoelectric effect [Catalan, G., Lubk, A., Vlooswijk, A. H. G., Snoeck, E. Magen, C., Janssens, A., Rispens, G. Rijnder, G., Blank, D. H. A., & Noheda, B. Flexoelectric rotation of polarization in ferroelectric thin films. *Nat. Mater.* **10**, 963-967 (2011)]. The broad $(A1/B1)_{(-110)}$ -type fine domain walls are distinct from the atomically sharp $(A1/A2)_{(100)}$ (or $(B1/B2)_{(010)}$) domain walls that are solely called fine domain walls in the original manuscript. One may be concerned about the effect of the junction between two types of constituent fine-domain walls within a composite wall. We believe this effect can be important at locations that are closer to or equal to the width of the sharp fine-domain wall from the connection point. The atomically sharp nature of fine domain walls leads us to conclude that the strain gradient is mainly attributed to the $(A1/B1)_{(-110)}$ -type intrinsic domain walls. We can find a region where fine domains are collapsed into a single large domain in TEM specimens while the macro-domain walls still endure the strain relaxation. In that case, the macro-domain walls along $\langle 110 \rangle$ consist of a single intrinsic wall, respectively, i.e. the macro-domain wall becomes a non-composite intrinsic wall that still shows a broad width.

We also consider the possibility of the $(A1/B2)$ -type (or $(A2/B1)$ -type) domain walls along $[110]$, which are expected to be W' walls according to the compatibility relation. The surfaces of these domain walls are not perpendicular to the substrate. The reason for forcing the W -type domain walls within the macro-domain wall is due to the mosaic rotations. The monoclinic unit-cells are rotated counter-clockwise or clockwise around x - or y -axis cooperatively so that the c -axis is normal to the substrate. As a result of the mosaic rotations, the ab -plane of unit cells are tilt from the substrate surface. The matching of the planes is necessary at the macro-domain walls, and thus the W -type walls are preferentially selected.

Spatial fluctuation in the centerline of a macro-domain wall (called domain wall roughness) can be induced by uneven widths of the fine-domains. If the horizontal width of $A1$ fine-domain is not equal to the vertical width of $B1$ fine-domain (Fig. R1c), the junction can deviate from the $y=x$ line. Although we admit the possibility, we still argue the existence of a strain-gradient region. One direct evidence is Fig. 3a. The wall centerline follows $y=x$ line in the low-magnification TEM image (in the inset), but we still observe the broad strain-gradient region around the domain wall.

The comparison with Kim et al basically OK now. In Kim et al. the wall thickness is more intrinsic because of the higher symmetry. What I like about the paper are two aspects. We now know that

1. it is possible to grow twinned WO_3 thin films. This is new.
2. the piezoelectricity is inversed with respect to the Kim et al paper. There the bulk is piezoelectric and the walls are not. Now the bulk is not piezoelectric but the walls are.

I find the fact that the measured walls are thick not terribly exciting because the thickness is basically an artefact of the wall roughness.

Nevertheless, the paper is interesting and can, from my perspective been accepted if a more thorough symmetry analysis and a discussion of the effect of the wall roughness are added.

Response:

We sincerely appreciate the reviewer's insightful comments and the positive evaluation once again. We are glad to have a chance to improve the manuscript based on the valuable comments regarding symmetry analysis and intrinsic twin walls.

List of changes

The modified parts in the main text are highlighted in red.

Fig. 1a and Fig. 2a are modified by including domain symbols "A1, A2, B1, B2".

Fig. 1f: the previous green circles are removed.

REVIEWERS' COMMENTS:

Reviewer #1 (Remarks to the Author):

I have no further comments. I think the current manuscript is acceptable for publication.

Reviewer #4 (Remarks to the Author):

The paper is much improved. I suggest acceptance.

REVIEWERS' COMMENTS:

Reviewer #1 (Remarks to the Author):

I have no further comments. I think the current manuscript is acceptable for publication.

Reviewer #4 (Remarks to the Author):

The paper is much improved. I suggest acceptance.

We thank the two reviewers for suggesting acceptance of our manuscript.